# A Comprehensive Graph Pooling Benchmark: Effectiveness, Robustness and Generalizability

## Abstract

Graph pooling has gained attention for its ability to obtain effective node and graph representations for various downstream tasks. Despite the recent surge in graph pooling approaches, there is a lack of standardized experimental settings and fair benchmarks to evaluate their performance. To address this issue, we have constructed a comprehensive benchmark that includes 17 graph pooling methods and 28 different graph datasets. This benchmark systematically assesses the performance of graph pooling methods in three dimensions, i.e., effectiveness, robustness, and generalizability. We first evaluate the performance of these graph pooling approaches across different tasks including graph classification, graph regression and node classification. Then, we investigate their performance under potential noise attacks and out-of-distribution shifts in real-world scenarios. We also involve detailed efficiency analysis, backbone analysis, parameter analysis and visualization to provide more evidence. Extensive experiments validate the strong capability and applicability of graph pooling approaches in various scenarios, which can provide valuable insights and guidance for deep geometric learning research. The source code of our benchmark is available at https://anonymous.4open.science/r/Graph_Pooling_Benchmark-8EDD.

## 1 Introduction

Recently, graph neural networks (GNNs) have garnered significant attention with extensive benchmarks (Tan et al., 2023; Li et al., 2024; Hu et al., 2020a) due to their remarkable ability to process graph-structured data across various domains including social networks (Wu et al., 2020a; Yang et al., 2021; Zhang et al., 2022), rumor detection (Bian et al., 2020; Zhang et al., 2023), biological networks (Wu et al., 2018; Choi et al., 2020), recommender systems (Ma et al., 2020a) and community detection (Alsentzer et al., 2020; Sun et al., 2022). Graph pooling approaches play a crucial role in GNNs by enabling the hierarchical reduction of graph representations, which is essential for capturing multi-scale structures and long-range dependencies (Liu et al., 2022a; Wu et al., 2022b; Dwivedi et al., 2023). They can preserve crucial topological semantics and relationships, which have shown effective for tasks including graph classification, node clustering, and graph generation (Liu et al., 2022a; 2020; Grattarola et al., 2022a; Li et al., 2024). In addition, by aggregating nodes and edges, graph pooling can also simplify large-scale graphs, facilitating the application of GNNs in real-world problems (Defferrard et al., 2016; Ying et al., 2018b; Mesquita et al., 2020; Zhang et al., 2020b; Tsitsulin et al., 2023b). Therefore, understanding and enhancing graph pooling approaches is the key to increasing GNN performance, driving the progress of deep geometric learning.

In literature, existing graph pooling approaches can be roughly divided into two categories (Bianchi & Lachi, 2024; Liu et al., 2022a), i.e., node dropping pooling (Knyazev et al., 2019; Lee et al., 2019; Ranjan et al., 2020; Ma et al., 2020b; Zhang et al., 2020a; Zhou et al., 2022; Pang et al., 2021; Bacciu et al., 2023; Zhang et al., 2020a; 2019; Song et al., 2024) and node clustering pooling approaches (Ying et al., 2018a; Bianchi et al., 2020; Duval & Malliaros, 2022; Wu et al., 2022a; Hansen & Bianchi, 2023; Tsitsulin et al., 2023a; Bianchi, 2022), based on the strategies used to simplify the graph. Node dropping pooling utilizes a learnable scoring function to guide the deletion of nodes with relatively low importance scores, resulting in lower computational costs, while node clustering pooling approaches typically treat graph pooling as a node clustering problem, where clusters are considered as new nodes for the coarsened graph (Liu et al., 2022b; Bianchi & Lachi, 2024). Even

though graph pooling research is becoming increasingly popular, there is still no standardized benchmark that allows for an impartial and consistent comparison of various graph pooling methods. Furthermore, due to the diversity and complexity of graph datasets, numerous experimental settings have been used in previous studies, such as varied proportions of training data and train/validation/test splits (Bian et al., 2020; Hansen & Bianchi, 2023; Dwivedi et al., 2023; Xu et al., 2024b). As a result, a comprehensive and publicly available benchmark of graph pooling approaches is highly expected that can facilitate the evaluation and comparison of different approaches, ensuring the reproducibility of results and further advancing the area of graph machine learning.

Towards this end, we present a comprehensive graph pooling benchmark, which includes 17 graph pooling methods and 28 datasets across different graph machine learning problems. In particular, we extensively investigate graph pooling approaches across three key perspectives, i.e., *effectiveness*, *robustness*, and *generalizability*. To begin, we provide a fair and thorough *effectiveness* comparison of existing graph pooling approaches across graph classification, graph regression and node classification. Then, we evaluate the *robustness* of graph pooling approaches under both noise attacks on graph structures and node attributes. In addition, we study the *generalizability* of different approaches under out-of-distribution shifts from both size and density levels. Finally, we include efficiency comparison, parameter analysis and backbone analysis for completeness.

Based on extensive experimental results, we have made the following observations: (1) Node clustering pooling methods outperform node dropping pooling methods in terms of robustness, generalizability, and performance on graph regression tasks. (2) Node clustering pooling methods incur higher computational costs, and both approaches exhibit comparable performance on graph classification tasks. (3) AsymCheegerCut-Pool and ParsPool demonstrate strong performance in graph classification tasks. (4) As the scale of graph data decreases, the performance gap between different graph pooling methods in node classification tasks increases, with KMISPool and ParsPool exhibiting outstanding performance. (5) Most graph pooling approaches experience significant performance degradation due to distribution shifts and are also challenged by class imbalance issues, but the extent of this impact varies across different datasets. (6) Node clustering pooling is relatively superior to node dropping pooling in terms of robustness and generalizability, while KMISPool demonstrates relatively better robustness and generalizability in node dropping pooling methods.

The main contributions of this paper are as follows:

- *Comprehensive Benchmark.* We present a comprehensive graph pooling benchmark, which incorporates 17 state-of-the-art graph pooling approaches and 28 diverse datasets across graph classification, graph regression, and node classification.

- *Extensive Analysis.* To investigate the pros and cons of graph pooling approaches, we thoroughly evaluate current approaches from three perspectives, i.e., *effectiveness*, *robustness*, and *generalizability*, which can serve as guidance for researchers in different applications.

- *Open-source Material.* We will make our benchmark of all these graph pooling approaches available and reproducible, and we believe our benchmark can benefit researchers in both graph machine learning and interdisciplinary fields.

## 2 Preliminaries

**Notations**. Consider a graph $G$ characterized by a vertex set $V$ and an edge set $E$. The features associated with each vertex are represented by the matrix $\boldsymbol{X} \in \mathbb{R}^{|V| \times d}$, where $|V|$ denotes the number of vertices, and $d$ signifies the dimensionality of the attribute vectors. The adjacency relationships within the graph are encapsulated by the adjacency matrix $\boldsymbol{A} \in \{0,1\}^{|V| \times |V|}$, where an entry $\boldsymbol{A}[i,j] = 1$ indicates the presence of an edge between vertex $v_i$ and vertex $v_j$; otherwise, $\boldsymbol{A}[i,j] = 0$.

**Graph Pooling** (Liu et al., 2022b; Grattarola et al., 2022b; Bianchi & Lachi, 2024). The aim of graph pooling is to reduce the spatial size of feature maps while preserving essential semantics, which thereby decreases computational complexity and memory usage. In this work, we focus on hierarchical pooling

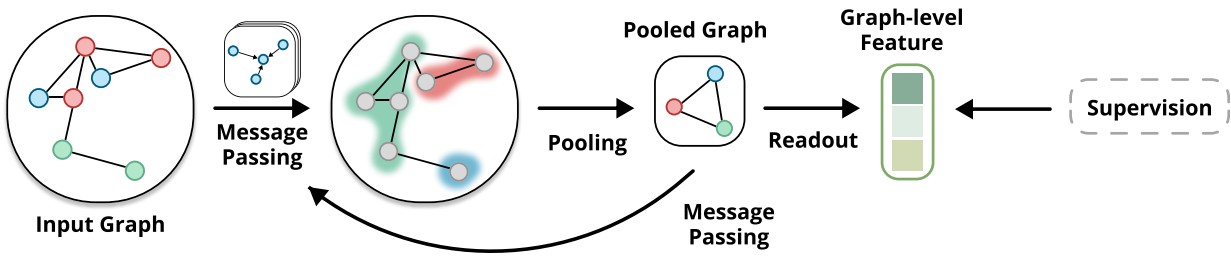

Figure 1: Overview of our hierarchical backbone for graph classification and graph regression.

approaches (Ying et al., 2018b). Let `POOL` denote a graph pooling function which maps $G$ to a graph $G' = (V', E')$ with the reduced size:

$$G' = \texttt{POOL}(G), \tag{1}$$

where $|V'| < |V|$. Current graph pooling methods can be classified into two categories, i.e., node dropping methods and node clustering methods (Bianchi & Lachi, 2024; Liu et al., 2022a). For node dropping methods, they focus on simplifying the graph by removing less important nodes and edges. It consists of three components, i.e., score generator `SCORE()` for computing the importance score for each node, node selector $\texttt{Select}_k()$ for selecting the top-$k$ nodes with the highest importance scores, and coarsing operator `COARSEN()` for generating a new coarsened graph based on `SCORE()` and $\texttt{Select}_k()$ Liu et al. (2022a). The process can be formulated as follows:

$$\mathbf{S}^{(l)} = \texttt{SCORE}(\mathbf{H}^{(l)}, \mathbf{A}^{(l)}), \quad \text{idx}^{(l+1)} = \texttt{Select}_k(\mathbf{S}^{(l)}), \quad \mathbf{H}^{(l+1)}, \mathbf{A}^{(l+1)} = \texttt{COARSEN}(\mathbf{H}^{(l)}, \mathbf{A}^{(l)}, \mathbf{S}^{(l)}, \text{idx}^{(l+1)}), \tag{2}$$

where $\mathbf{H}^{(l)}$ is the feature matrix and $\mathbf{A}^{(l)}$ is the adjacency matrix for layer $l$, $\mathbf{S}^{(l)} \in \mathbb{R}^{n \times 1}$ represents the significance scores, $\texttt{SELECT}_k()$ ranks the scores and returns the indices of the top-$k$ values in $\mathbf{S}^{(l)}$, and $\text{idx}^{(l+1)}$ denotes the indices of the reserved nodes.

For node clustering methods, they focus on grouping nodes in the graph into several clusters, where each cluster represents a supernode. The graph is then simplified based on these supernodes. It consists of three components, i.e., the cluster assignment operator `CAM()`, which defines how nodes in the original graph are assigned to clusters in the pooled graph, mapping the nodes to the coarsened supernodes and coarsing operator `COARSEN()` to generate a new coarsened graph based on `CAM()`. The process can be formulated as follows:

$$\mathbf{C}^{(l)} = \texttt{CAM}(\mathbf{H}^{(l)}, \mathbf{A}^{(l)}), \quad \mathbf{H}^{(l+1)}, \mathbf{A}^{(l+1)} = \texttt{COARSEN}(\mathbf{H}^{(l)}, \mathbf{A}^{(l)}, \mathbf{C}^{(l)}). \tag{3}$$

where $C^{(l)} \in \mathbb{R}^{n_l \times n_{l+1}}$ represents the learned cluster assignment matrix.

**Graph Classification and Regression** (Knyazev et al., 2019; Chen et al., 2019; Grattarola et al., 2022b). The two primary graph-level tasks are graph regression and graph classification. Here, a graph dataset $\mathcal{G}$ is provided as a set of graph-label pairs $(G_i, y_i)$, where $y_i$ denotes the label for graph $G_i$. The objective is to train a powerful discriminative model $f$ that predicts the correct label $y_i$ given an input graph $G_i$. In graph classification, $y_i$ are categorical labels $1, \cdots, K$ with $K$ as the number of classes, while in graph regression, $y_i$ are continuous values. A well-trained graph classification model should output labels that closely match the true labels, and similarly, a graph regression model should predict values that are nearly identical to the ground truth values. In these tasks, graph pooling always accompanies graph convolutional operators. In formulation, the basic updating rule is written as follows:

$$\boldsymbol{H}^{(l+1)} = \sigma(\tilde{\boldsymbol{D}}^{-\frac{1}{2}} \tilde{\boldsymbol{A}} \tilde{\boldsymbol{D}}^{-\frac{1}{2}} \boldsymbol{H}^{(l)} \boldsymbol{W}^{(l)}), \tag{4}$$

where $\boldsymbol{H}^{(l)}$ denotes the node feature matrix at layer $l$, $\boldsymbol{W}^{(l)}$ denotes the weight matrix at the corresponding layer, $\tilde{\boldsymbol{A}} = \boldsymbol{A} + \boldsymbol{I}$ is the adjacency matrix $\boldsymbol{A}$ plus the identity matrix $\boldsymbol{I}$, $\tilde{\boldsymbol{D}}$ is the degree matrix of $\tilde{\boldsymbol{A}}$, and $\sigma$ is a nonlinear activation function (Kipf & Welling, 2016). The pooling layers can be formulated as:

$$\boldsymbol{H}^{(pool)} = \texttt{POOL}(\boldsymbol{H}^{(L)}), \tag{5}$$

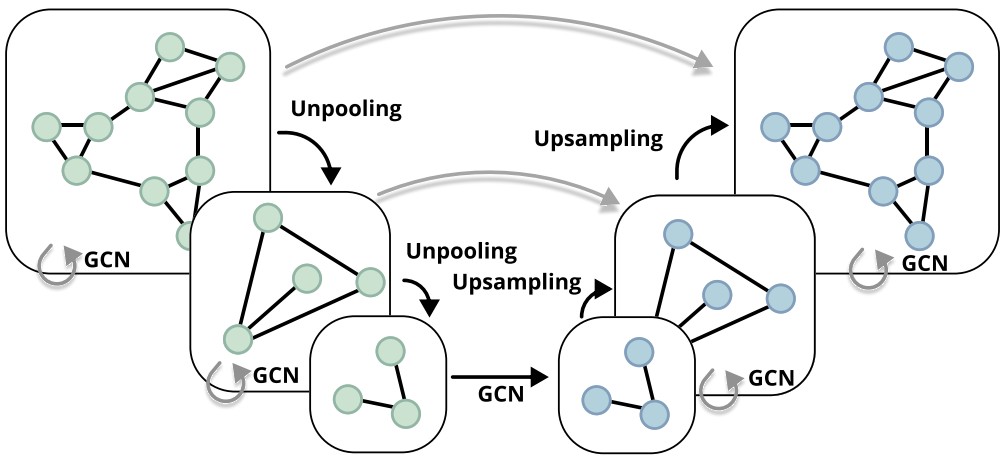

Figure 2: Overview of the graph U-Net framework for node classification.

where $\boldsymbol{H}^{(pool)}$ is the node feature matrix after pooling We iteratively conduct graph convolution and graph pooling operators and adopt a readout function to output the graph representation for downstream tasks. The overview of the basic hierarchical backbone can be found in Figure 1.

**Node Classification** (Kipf & Welling, 2016; Veličković et al., 2018; Perozzi et al., 2014). The aim of node classification is to assign semantic labels to nodes in a graph according to their attributes and relationships with different nodes. Each dataset involves a graph $G$, consisting of nodes $v_i$ and their corresponding labels $y_i$. $|V|$ is divided into a labeled set $V^l$ and a unlabeled set $V^u$. We are required to train a graph neural network model that can predict the missing labels of nodes in $V^u$ using the attributes of other nodes. U-Net framework (Ronneberger et al., 2015) is widely used to incorporate pooling operations for node classification (Wu et al., 2022a; Song et al., 2024; Zhang et al., 2021). Figure 2 shows the overview of the U-Net framework. In the encoder part, U-Net progressively applies pooling and graph convolution to downsample the graphs and extract multi-scale features. The decoder part of U-Net utilizes upsampling and graph convolution to gradually upsample the low-resolution feature maps back to the original graph size. Residual connections are employed to directly transfer the feature maps from the encoder to the decoder, facilitating the preservation of fine-grained semantics during upsampling (Ronneberger et al., 2015; Ibtehaz & Rahman, 2020; Leng et al., 2018). Pooling not only simplifies the graph's complexity but also provides the model with multi-scale feature representation capabilities.

In particular, in the downsampling path, the input feature matrix is first subjected to graph convolution, where the product of the adjacency matrix and the feature matrix, along with the weighted sum of the weight matrix and the bias term, yields the activated feature matrix $\mathbf{H}^{(l+1)}$. Next, a pooling operation is applied, reducing the number of nodes by selecting those with higher scores, thereby transforming the original feature matrix $\mathbf{H}^{(l+1)}$ and adjacency matrix $\mathbf{A}^{(l)}$ into a smaller feature matrix $\mathbf{H}'^{(l+1)}$ and a corresponding adjacency matrix $\mathbf{A}'^{(l+1)}$. In the upsampling path, the pooled feature matrix $\mathbf{H}'^{(l+1)}$ is first upsampled to restore the original number of nodes, generating a new feature matrix $\mathbf{H}''^{(l+1)}$. Then, the restored feature matrix is concatenated with the corresponding feature matrix $\mathbf{H}^{(l)}$ from the downsampling path, forming the merged feature matrix $\mathbf{H}^{(l+1)}_{\mathrm{merged}}$. Finally, the merged feature matrix undergoes another graph convolution, resulting in the output feature matrix $\mathbf{H}^{(l+2)}$.

# 3 Graph Pooling Benchmark

## 3.1 Graph Pooling Approaches

Graph pooling plays a significant role in graph mining learning. First, its primary function is to compress node representations into a smaller graph or a single vector, which can reduce computational complexity.

This is particularly important for processing large-scale graph data with scalability (Knyazev et al., 2019; Ju et al., 2024). Second, graph pooling generates global graph representations by aggregating local node information, enabling better capture of substructures and motifs (Song et al., 2024). Third, hierarchical pooling methods, such as node clustering pooling and node drop pooling, coarsen the graph step-by-step, extracting features at multiple levels. This helps capture complex structures and multi-level information, thereby enhancing the model's expressive power (Zhou et al., 2022; Bacciu et al., 2023). Finally, graph pooling improves the model's generalization ability to unseen graph data by reducing graph complexity and noise, enhancing robustness, especially in out-of-distribution (OOD) scenarios (Duval & Malliaros, 2022; Ju et al., 2024). Therefore, we introduce this benchmark for evaluating and understand graph pooling approaches.

Our benchmark contains 17 state-of-the-art graph pooling approaches (detailed in Table 1): Top-KPool (Knyazev et al., 2019), SAGPool (Lee et al., 2019), ASAPool (Ranjan et al., 2020), PANPool (Ma et al., 2020b), COPool (Zhou et al., 2022), CGIPool (Pang et al., 2021), KMISPool (Bacciu et al., 2023), GSAPool (Zhang et al., 2020a), HGPSLPool (Zhang et al., 2019), AsymCheegerCutPool (Hansen & Bianchi, 2023), DiffPool (Ying et al., 2018a), MincutPool (Bianchi et al., 2020), DMoNPool (Tsitsulin et al., 2023a), HoscPool (Duval & Malliaros, 2022), JustBalancePool (Bianchi, 2022), SEPool (Wu et al., 2022a), and ParsPool (Song et al., 2024). Then, we introduce the details of these graph pooling approaches:

- **TopKPool (Knyazev et al., 2019).** TopKPool utilizes the attention mechanism to learn the scores of different nodes and then selects the nodes with top scores, which can learn important local portions from original graphs.

- **SAGPool (Lee et al., 2019).** SAGPool utilizes a different graph neural network to learn importance scores, which can guide the pooling process effectively.

- **ASAPool (Ranjan et al., 2020).** ASAPool considers the neighboring subgraphs to represent nodes and then adopts the attention mechanism to generate subgraph representations. The importance nodes are selected by a graph neural networks with local extremum information.

- **PANPool (Ma et al., 2020b).** PANPool constructs the maximal entropy transition (MET) matrix based on graph Laplacian, which can generate importance scores for different nodes.

- **COPool (Zhou et al., 2022).** COPool learns pooled representations from the complimentary edge and node views. The edge view comes from high-order semantics information while the node view stems from importance scores from the cut proximity matrix.

- **CGIPool (Pang et al., 2021).** CGIPool incorporates mutual information optimization into graph pooling, which can enhance the graph-level relationships between the original graph and the pooled graph.

- **KMISPool (Bacciu et al., 2023).** KMISPool incorporates the Maximal k-Independent Sets (k-MIS) into graph pooling, which can detect the important nodes in the graph with topological preserved.

- **GSAPool (Zhang et al., 2020a).** GSAPool integrates both structural and attribute information in different components. These scores from different components are then fused to guide the graph pooling process.

- **HGPSLPool (Zhang et al., 2019).** HGPSLPool not only utilizes graph pooling to determine important nodes from the original graph, but also leverages graph structure learning to explore the topological information in the pooled graph.

- **AsymCheegerCutPool (Hansen & Bianchi, 2023).** AsymCheegerCutPool conducts graph clustering to generate the assignment of each node according to graph total variation (GTV). Each cluster is aggregated in a hierarchical manner during graph pooling.

- **DiffPool (Ying et al., 2018a).** DiffPool introduces a learnable soft assignment of each node during graph clustering, and then maps each cluster into the coarsened nodes in the pooling graph.

Table 1: Overview of experimental details of graph pooling research. These papers utilize different settings, which validates the necessity of building a comprehensive and fair benchmark.

| Methods | | Datasets | Tasks |
|---|---|---|---|
| *Node Dropping Pooling* | | | |
| TopKPool | NIPS'19 | MNIST, COLLAB, PROTEINS, D&D | Graph Classification |
| SAGPool | ICML'19 | D&D, PROTEINS, NCI1, NCI109, FRANKENSTEIN | Graph Classification |
| ASAPool | AAAI'20 | D&D, PROTEINS, NCI1, NCI109, FRANKENSTEIN | Graph Classification |
| PANPool | NIPS'20 | PROTEINS, PROTEINS_FULL, NCI1, AIDS, MUTAGENCITY | Graph Classification |
| COPool | ECMLPKDD'22 | BZR, AIDS, NCI1, NCI109, PROTEINS, QM7, IMDB-M | Graph Classification, Graph Regression |
| CGIPool | SIGIR'22 | NCI1, NCI109, MUTAG, IMDB-B, IMDB-M, COLLAB, PROTEINS | Graph Classification |
| KMISPool | AAAI'23 | D&D, REDDIT-B, REDDIT-5K, REDDIT-12K, Github | Graph Classification, Node Classification |
| GSAPool | WWW'20 | D&D, NCI1, NCI109, MUTAG | Graph Classification |
| HGPSLPool | Arxiv'19 | D&D, PROTEINS, NCI1, NCI109, ENZYMES, MUTAG | Graph Classification |
| *Node Clustering Pooling* | | | |
| AsymCheegerCutPool | ICML'23 | Cora, Citeseer, Pubmed, DBLP | Node Classification |
| DiffPool | NIPS'18 | D&D, PROTEINS, COLLAB, ENZYMES, REDDIT-MULTI | Graph Classification |
| MincutPool | ICML'20 | D&D, PROTEINS, COLLAB, REDDIT-B, MUTAG, QM9 | Graph Classification, Graph Regression |
| DMoNPool | JMLR'23 | Cora, Citeseer Pubmed, Coauthor | Node Classification |
| HoscPool | CIKM'22 | Cora, Citeseer Pubmed, Coauthor, DBLP, Email-EU | Node Classification |
| JustBalancePool | Arxiv'22 | Cora, Citeseer, Pubmed, DBLP | Node Classification |
| SEPool | ICML'22 | IMDB-B, IMDB-M, COLLAB, MUTAG, Cora, Citeseer, Pubmed | Graph Classification, Node Classification |
| ParsPool | ICLR'24 | D&D, PROTEINS, NCI1, NCI109, Ogbg-molpcba, Cora, Citeseer, Pubmed | Graph Classification, Node Classification |

- **MincutPool (Bianchi et al., 2020).** MincutPool relaxes the classic normalized mincut problem into a continuous fashion, and then optimizes a graph neural network to achieve this. The graph clustering results are adopted to guide the graph pooling process.

- **DMoNPool (Tsitsulin et al., 2023a).** DMoNPool introduces an objective based on modularity for graph clustering and then adds a regularization term to avoid trivial solutions during optimization. Similarly, graph clustering results are leveraged for graph pooling.

- **HoscPool (Duval & Malliaros, 2022).** HoscPool combines higher-order relationships in the graph with graph pooling based on motif conductance. It minimizes a relaxed motif spectral clustering objective and involves multiple motifs to learn hierarchical semantics.

- **JustBalancePool (Bianchi, 2022).** JustBalancePool consists of two components. On the one hand, it aims to reduce the local quadratic variation during graph clustering. On the other hand, it involves a balanced term to reduce the risk of degenerate solutions.

- **SEPool (Wu et al., 2022a).** SEPool generates a clustering assignment matrix in one go through a global optimization algorithm, avoiding the suboptimality associated with layer-by-layer pooling.

- **ParsPool (Song et al., 2024).** ParsPool is characterized by the introduction of a graph parsing algorithm that adaptively learns a personalized pooling structure for each graph. ParsPool is inspired by bottom-up grammar induction and can generate a flexible pooling tree structure for each graph.

## 3.2 Datasets

To systematically evaluate graph pooling methods, we integrate 28 datasets from different domains. For graph classification, we select eleven publicly available datasets from TUDataset (Morris et al., 2020), including seven molecules datasets, i.e., MUTAG (Debnath et al., 1991), NCI1 (Wale et al., 2008), NCI109 (Wale et al., 2008), COX2 (Sutherland et al., 2003), AIDS (Riesen & Bunke, 2008), FRANKENSTEIN (Orsini et al., 2015), and Mutagenicity (Debnath et al., 1991), four bioinformatics datasets, i.e. D&D (Shervashidze et al., 2011), PROTEINS (Borgwardt et al., 2005), PROTEINS_FULL (Borgwardt et al., 2005), and EN-ZYMES (Schomburg et al., 2004), three social network dataset, i.e., IMDB-BINARY (IMDB-B) (Cai & Wang, 2018), IMDB-MULTI (IMDB-M) (Cai & Wang, 2018), and COLLAB (Cai & Wang, 2018). We also include a large-scale graph classification dataset, Ogbg-molpcba, from the Open Graph Benchmark (OGB) (Hu et al., 2020b). For graph regression, we choose six datasets from MoleculeNet (Wu et al., 2018) including QM7, QM8, BACE, ESOL, FreeSolv, and Lipophilicity. For node classification, we utilize three citation networks, i.e., Cora, Citeseer, and Pubmed (Yang et al., 2016), three website networks, i.e., Cornell, Texas, and Wisconsin (Pei et al., 2020), and the GitHub dataset (Rozemberczki et al., 2021). We also obtain a large-scale dataset, Ogbn-arxiv, from OGB (Hu et al., 2020b). More information about the summary statistics and description of the datasets are detailed in the Appendix A.

Table 2: Results of **graph classification** for different graph pooling methods. The mean and variance of average precision (Ogbg-molpcba) and accuracy (remaining datasets) are reported. The best and 2nd best are noted in bold font and underlined, respectively. OOM denotes out of GPU memory, and OOT denotes cannot be computed within 24 hours.

| Methods | Ogbg-molpcba | PROTEINS | NCI1 | NCI109 | MUTAG | D&D | IMDB-B | IMDB-M | COLLAB | Avg. | Rank |
|---|---|---|---|---|---|---|---|---|---|---|---|
| *Node Drop Pooling* | | | | | | | | | | | |
| TopKPool | 15.79±0.42 | 70.83±1.25 | 70.34±1.80 | 69.65±1.61 | 82.76±4.88 | 69.07±5.52 | 74.44±3.71 | 48.44±3.46 | 75.38±1.13 | 70.11 | 12.44 |
| SAGPool | 21.08±2.19 | 74.64±1.53 | 73.10±1.21 | 71.29±0.82 | 81.61±5.86 | 73.27±1.12 | 75.33±3.31 | 48.74±3.09 | 77.91±2.22 | 71.99 | 8.06 |
| ASAPool | OOT | 73.69±1.48 | 73.48±1.03 | 70.45±0.84 | 72.41±10.15 | OOT | 71.56±3.46 | 46.96±3.72 | OOT | 68.09 | 13.56 |
| PANPool | 21.81±0.99 | 70.60±1.67 | 73.29±1.07 | 70.84±1.23 | 78.16±8.60 | 73.27±4.05 | 73.33±3.57 | 47.70±3.58 | 78.40±2.80 | 70.70 | 11.00 |
| COPool | 25.50±2.46 | **75.24±2.46** | 74.10±1.06 | 71.35±1.05 | 83.91±3.25 | 73.57±0.42 | 74.44±4.40 | 48.89±3.82 | 81.33±1.15 | 72.85 | 5.28 |
| CGIPool | 23.78±6.71 | 73.57±1.49 | 75.72±1.65 | 73.81±0.42 | 86.21±4.88 | 72.07±1.47 | 74.22±3.62 | 46.22±2.02 | 80.40±1.71 | 72.78 | 8.22 |
| KMISPool | 26.85±0.28 | 70.63±1.01 | 73.15±2.19 | 73.17±1.10 | 80.46±4.30 | 70.57±1.70 | 72.89±3.62 | 46.96±2.47 | 80.71±0.49 | 71.07 | 9.83 |
| GSAPool | **26.95±1.36** | 72.14±1.09 | 71.12±1.33 | 70.65±1.45 | 87.36±1.63 | 72.97±1.27 | 74.67±3.93 | 46.37±4.13 | 76.84±2.11 | 71.52 | 9.28 |
| HGPSLPool | 22.78±0.51 | 72.02±1.73 | 72.22±0.42 | 70.35±1.31 | 71.26±12.70 | 73.27±2.78 | 72.89±4.37 | 46.81±2.19 | 79.24±0.80 | 69.76 | 12.17 |
| *Node Clustering Pooling* | | | | | | | | | | | |
| AsymCheegerCutPool | 24.82±0.60 | 74.60±1.96 | 75.90±1.69 | 73.98±1.88 | **89.66±2.82** | 74.47±2.36 | 74.89±3.14 | 48.30±3.63 | 80.62±1.06 | **74.05** | 4.72 |
| DiffPool | 25.21±0.42 | 74.80±1.71 | 74.72±1.82 | 75.16±0.35 | 80.46±5.86 | 73.57±1.85 | 74.44±0.63 | 47.70±4.01 | 78.89±0.55 | 72.47 | 6.50 |
| MincutPool | 24.97±0.41 | 72.42±1.71 | 75.53±1.15 | 74.30±1.33 | 85.06±1.63 | 71.77±2.12 | 73.78±3.94 | 45.93±2.55 | 76.53±1.60 | 71.91 | 9.56 |
| DMoNPool | 24.75±0.67 | 68.45±4.79 | 72.45±0.15 | 71.18±1.66 | 75.86±5.63 | 75.38±0.42 | 73.33±3.81 | 47.26±1.68 | 77.07±0.44 | 70.12 | 11.17 |
| HoscPool | 24.63±0.37 | 72.42±0.74 | 76.88±0.61 | 76.13±1.97 | 85.06±1.63 | 71.77±2.12 | 74.67±1.96 | 45.93±2.77 | 78.18±1.69 | 72.63 | 8.06 |
| JustBalancePool | 25.19±0.42 | 68.85±2.97 | 76.34±0.46 | **76.34±1.51** | 81.61±1.63 | 71.77±2.12 | 74.89±4.09 | 45.93±5.08 | 77.87±1.26 | 71.70 | 8.67 |
| SEPool | OOT | 62.25±4.57 | 62.77±2.25 | 63.74±2.30 | 67.22±8.41 | 80.26±3.04 | 77.00±4.05 | 54.13±3.71 | 75.64±2.04 | 67.88 | 11.39 |
| ParsPool | 26.63±0.30 | 75.02±0.64 | **77.07±0.23** | 76.20±0.44 | 79.31±5.63 | 76.10±0.80 | 75.11±2.20 | 49.48±0.91 | **83.60±0.50** | 73.99 | **3.11** |

### 3.3 Evaluation Protocols

Our benchmark evaluation encompasses three key aspects, i.e., effectiveness, robustness, and generalizability. We perform a hyperparameter search for all pooling methods; detailed information can be found in Appendix B. *Firstly*, we conduct a performance comparison of graph pooling approaches across three tasks including graph classification, graph regression, and node classification. For graph and node classification tasks, we employ average precision for Ogbg-molpcba, and accuracy for remaining datasets as the evaluation metric. For graph regression, we use root mean square error (RMSE) for ESOL, FreeSolv, and Lipophilicity (Wu et al., 2018). Following previous research (Xu et al., 2024b), we use the area under the receiver operating characteristic (AUROC) curve to evaluate BACE, and mean absolute error (MAE) for QM7 and QM8. *Secondly*, our benchmark evaluates the robustness of graph pooling approaches in both graph-level and node-level tasks across two perspectives: structural robustness and feature robustness (Li & Wang, 2018). In particular, we add and drop edges of graphs to study structural robustness and mask node features to investigate feature robustness. *Thirdly*, we employ size-based and density-based distribution shifts to evaluate the generalizability of different pooling methods in graph-level tasks under real-world scenarios (Gui et al., 2022). We also use degree-based and closeness-based distribution shifts to assess the generalizability of different pooling methods in node-level tasks. In addition to these three views, we conduct a further analysis of these graph pooling approaches including the comparison of efficiency, visualization, and different backbone parameter choices.

## 4 Experiment

### 4.1 Experimental Settings

All graph pooling methods in our benchmark are implemented by PyTorch (Paszke et al., 2019). Graph convolutional networks serve as the default encoders for all algorithms. The experimental setup includes a Linux server equipped with NVIDIA A100 GPUs, with an Intel Xeon Gold 6354 CPU. The software stack comprises PyTorch 1.11.0, PyTorch-geometric 2.1.0 (Fey & Lenssen, 2019), and Python 3.9.16. More details about experimental settings can be found in Appendix B.

### 4.2 Effectiveness Analysis

**Performance on Graph Classification.** To begin, we investigate the performance of different graph pooling approaches on graph classification. The results of compared approaches on seven popular datasets are recorded in Table 2. *Firstly*, in general, ParsPool, AsymCheegerCutPool, and COPool are the three best-

Table 3: Results of **graph regression** for different pooling methods. The mean and variance of MAE (QM7, QM8), AUROC (BACE), RMSE (ESOL, FreeSolv, Lipophilicity) are reported. - denotes cannot converge.

| Methods | QM7 | QM8 | BACE | ESOL | FreeSolv | Lipophilicity | Rank |
|---|---|---|---|---|---|---|---|
| *Node Drop Pooling* | | | | | | | |
| TopKPool | 63.39±9.66 | 0.021±0.001 | **0.85±0.02** | 0.96±0.06 | 1.92±0.37 | 0.80±0.02 | 8.1 |
| SAGPool | 97.69±11.19 | 0.023±0.001 | 0.84±0.01 | 1.16±0.07 | 2.31±0.66 | 0.93±0.06 | 13.1 |
| ASAPool | 56.79±6.17 | 0.029±0.008 | **0.85±0.02** | 0.92±0.03 | 1.92±0.37 | 0.78±0.05 | 8.2 |
| PANPool | 53.04±1.20 | **0.015±0.000** | 0.83±0.02 | 1.01±0.03 | 1.80±0.10 | 0.84±0.01 | 9.7 |
| COPool | 84.22±3.28 | 0.020±0.001 | **0.85±0.01** | 0.98±0.07 | 1.85±0.24 | 0.85±0.02 | 8.5 |
| CGIPool | 97.41±16.25 | 0.020±0.001 | 0.84±0.03 | 1.59±0.62 | 2.49±0.97 | 0.83±0.07 | 11.7 |
| KMISPool | 80.51±21.34 | 0.017±0.001 | **0.85±0.02** | 0.95±0.04 | 1.29±0.18 | 0.81±0.03 | 5.9 |
| GSAPool | 106.72±22.90 | 0.021±0.001 | **0.85±0.02** | 0.96±0.08 | 1.95±0.26 | 0.82±0.03 | 8.8 |
| HGPSLPool | **47.88±0.83** | **0.015±0.000** | 0.84±0.01 | 1.02±0.06 | 1.62±0.09 | 0.76±0.01 | 7.8 |
| *Node Clustering Pooling* | | | | | | | |
| AsymCheegerCutPool | 64.91±8.30 | 0.031 ± 0.005 | 0.84 ± 0.01 | 0.99 ± 0.12 | 2.00 ± 0.18 | 0.95 ± 0.11 | 12.9 |
| DiffPool | 54.98±3.44 | 0.037 ± 0.010 | 0.84 ± 0.02 | 0.81 ± 0.05 | 1.20 ± 0.09 | 0.73 ± 0.03 | 8.0 |
| MincutPool | - | 0.020 ± 0.001 | **0.85 ± 0.02** | 0.76 ± 0.02 | 1.19 ± 0.18 | 0.73 ± 0.02 | 4.3 |
| DMoNPool | - | 0.021 ± 0.001 | **0.85 ± 0.02** | **0.68 ± 0.02** | 1.16 ± 0.15 | **0.69 ± 0.02** | **3.5** |
| HoscPool | 59.44±21.48 | 0.019 ± 0.002 | 0.84 ± 0.01 | 0.76 ± 0.02 | **1.14 ± 0.13** | 0.72 ± 0.02 | 4.6 |
| JustBalancePool | - | 0.022 ± 0.004 | **0.85 ± 0.02** | 0.74 ± 0.03 | 1.26 ± 0.16 | 0.70 ± 0.01 | 4.9 |

performing pooling models, and the performance of all 17 pooling methods varies significantly across different datasets. No single pooling method consistently outperforms the others across all datasets. *Secondly*, it is noteworthy that SEPool achieves significant advantages on D&D, IMDB-B, and IMDB-M. This is because SEPool's coding tree structure, and these datasets are characterized by high clustering coefficients (Watts & Strogatz, 1998), which benefits the coding tree method because the hierarchical nature of the tree can better capture and represent these localized, highly connected substructures (Wu et al., 2022a). However, SEPool also implies greater computational resource overhead, which presents challenges when processing large-scale graph data such as Ogbg-molpcba. *Thirdly*, the methods with the highest average accuracy are ParsPool and AsymCheegerCutPool. ParsPool can capture a personalized pooling structure for each individual graph, while AsymCheegerCutPool calculates cluster assignments based on a tighter relaxation in terms of Graph Total Variation (GTV) (Song et al., 2024; Hansen & Bianchi, 2023). These methods are flexible, and the datasets differ significantly in diameter, degree, and clustering coefficients. *Fourthly*, GSAPool demonstrates superior performance on datasets with small-scale graphs, such as Ogbg-molpcba (Avg. nodes=26.0, Avg. edges=27.5) and MUTAG (Avg. nodes=17.9, Avg. edges=39.6). GSAPool combines structural information (SBTL) and node feature information (FBTL) to generate the final pooling topology. In terms of structural information, SBTL employs GCNConv, while for feature information, FBTL directly processes node features using MLP. GCNConv accurately identifies critical structural nodes (e.g., high-degree hub nodes) through neighborhood aggregation. The limited number of nodes enhances the distinctiveness of individual node features (e.g., atomic attributes in molecular graphs), enabling MLP to efficiently extract their importance. Additionally, on small-scale graphs, structural information (e.g., node degree and adjacency relations) and feature information (e.g., node attributes) are typically more pronounced and easier to capture, leading to better performance. ParsPool excels on datasets with large-scale graphs, such as D&D (Avg. nodes=284.3, Avg. edges=715.7) and COLLAB (Avg. nodes=74.5, Avg. edges=2457.2). The algorithm combines graph topology with continuous feature values through a learnable edge score matrix, generating clusters via three operations: DOM, EXP, and GEN. DOM selects key edges to anchor initial cluster centers, EXP dynamically absorbs neighboring nodes through expansion, and GEN produces a soft assignment matrix. This process adaptively partitions clusters based on local structures and global connectivity patterns without predefined cluster numbers or pooling ratios. For large-scale graphs with dense edges (e.g., COLLAB), the learnable edge scores enable gradient-based optimization to identify high-order semantic clusters while sparsifying non-critical edges. For graphs with numerous nodes (e.g., D&D), the row-wise max-indexing in DOM and neighborhood expansion in EXP efficiently aggregate nodes with linear complexity, avoiding the performance bottlenecks of traditional global clustering methods.

Table 4: Results of **node classification** for different pooling methods. No Pooling denotes without pooling layers.

| Methods | Ogbn-arxiv | Cora | Citeseer | Pubmed | Cornell | Texas | Wisconsin | Github | Avg. | Rank |
|---|---|---|---|---|---|---|---|---|---|---|
| TopKPool | 53.36±0.03 | 88.91±0.93 | 77.56±0.85 | 86.13±0.34 | 49.09±2.57 | 54.18±4.80 | 51.58±3.05 | 86.95±0.20 | 67.91 | 8.00 |
| SAGPool | 53.39±0.02 | 89.18±0.65 | 77.56±0.81 | 86.07±0.70 | 81.09±3.74 | 55.64±2.95 | 51.32±3.53 | 86.99±0.22 | 73.48 | 5.33 |
| ASAPool | OOM | 89.10±0.86 | 77.76±1.01 | 85.74±0.18 | 79.64±2.91 | 54.55±4.74 | 50.79±3.49 | OOM | 72.93 | 6.83 |
| PANPool | OOM | 89.05±0.96 | 77.20±0.98 | 85.88±0.11 | 78.91±2.47 | 56.36±5.14 | 50.53±3.18 | OOM | 72.99 | 8.00 |
| COPool | OOM | 89.00±0.70 | 77.26±0.89 | 85.27±0.27 | 77.82±3.13 | 56.36±5.98 | 52.37±2.68 | 86.68±0.20 | 73.01 | 7.50 |
| CGIPool | 53.60±0.39 | 89.15±0.84 | 77.40±0.81 | 85.92±0.66 | **81.82±4.30** | 54.55±4.15 | 51.05±3.85 | 86.88±0.22 | 73.32 | 6.33 |
| KMISPool | 53.38±0.08 | **89.74±0.02** | 77.75±0.01 | 87.80±0.01 | 79.56±1.31 | 81.42±1.64 | 82.31±0.50 | 87.09±0.04 | 83.10 | **2.50** |
| GSAPool | **53.76±0.11** | 89.05±0.77 | 77.16±0.92 | 86.21±0.73 | 80.36±3.71 | 54.18±3.88 | 51.84±4.29 | **87.12±0.09** | 73.13 | 6.67 |
| HGPSLPool | OOM | 89.08±0.83 | **77.84±0.67** | OOM | 58.55±3.71 | 55.27±3.37 | 51.58±1.75 | OOM | 69.71 | 6.33 |
| SEPool | OOT | 83.17±0.00 | 70.47±0.00 | 79.33±0.80 | 51.35±0.00 | 66.66±6.50 | 57.51±0.85 | OOT | 68.08 | 8.67 |
| ParsPool | OOM | 84.51±0.35 | 74.27±0.38 | **89.20±0.00** | 72.07±6.49 | **81.98±6.49** | 82.35±17.94 | OOM | 80.73 | 5.50 |
| No Pooling | 53.61±0.07 | 89.48±0.27 | 77.69±0.24 | 86.10±0.06 | 48.83±1.24 | 56.50±1.11 | 54.43±1.54 | 86.46±0.02 | 68.84 | 5.17 |

**Performance on Graph Regression.** We further explore the performance of different pooling methods through graph-level regression tasks. As shown in Table 3, we can observe that: *Firstly*, overall, node clustering pooling methods outperform node dropping pooling methods, with DMoNPool and MincutPool showing the best performance. The possible reason is that in graph regression tasks, the model's objective is to predict a continuous numerical output. Such tasks typically require capturing global structural features and continuity within the graph. Compared to node clustering pooling, node dropping pooling tends to lose more global information (Tsitsulin et al., 2023a; Bianchi et al., 2020). DMoNPool and MincutPool are more inclined to maintain the global characteristics of the graph rather than emphasizing the representation of locally important structures, which may result in their performance being inferior to that of ParsPool, AsymCheegerCutPool, and COPool in graph classification tasks. *Secondly*, in the BACE dataset, the performance of most pooling methods tends to be consistent, whereas in other datasets, there is a greater variance in performance. The possible reason is that although the graphs in the BACE dataset are relatively large, the average diameter is relatively small, so different pooling methods face fewer challenges in summarizing the global structural information of the graphs, which may lead to more consistent performance.

**Performance on Node Classification.** Table 4 presents the performance of various pooling methods in node classification tasks. We observe the following: *Firstly*, KMISPool and ParsPool demonstrate the best overall performance, significantly outperforming other methods on small-scale datasets such as Cornell, Texas, and Wisconsin. *Secondly*, node classification models without pooling layers achieve comparable results to most pooling methods across the majority of datasets. A potential reason for this is that pooling operations tend to lose substantial node information, which consequently weakens performance in node classification tasks. *Thirdly*, the scalability of ASAPool, PANPool, HGPSLPool, SEPool, and ParsPool still requires improvement, as they face memory/runtime bottlenecks, making it cannot complete training on larger datasets such as Ogbn-arxiv or GitHub. *Fourthly*, for datasets with strong connectivity between nodes, such as Ogbn-arxiv (Avg. degree=13.7) and Github (Avg. degree=15.3), GSAPool performs exceptionally well due to its emphasis on extracting structural information through GCNConv. Conversely, for datasets with weaker connectivity, such as PubMed (Avg. degree=4.5), Texas (Avg. degree=3.2), and Wisconsin (Avg. degree=3.7), KMISPool excels. KMISPool demonstrates superior performance on the Cora dataset, which has the highest average clustering coefficient (Avg. CC=0.24). A high clustering coefficient indicates strong community structure among nodes. The k-MIS algorithm selects nodes that are more than k hops apart as centroids, ensuring their uniform distribution across the graph and preventing over-sampling or omission in specific regions. This uniform node selection strategy is particularly crucial in graphs with weaker connectivity, where traditional pooling methods may suffer from information loss or uneven sampling due to insufficient local connections.

### 4.3 Robustness Analysis

The compared performance for three types of random noise on eight graph pooling methods on the PROTEINS, NCI1, NCI109, and MUTAG datasets are shown in Table 5. With a probability of 50%, edges of the graph are randomly removed or added, and node features are randomly masked. We can observe that: *Firstly*, overall, node clustering pooling demonstrates better robustness against three types of attacks

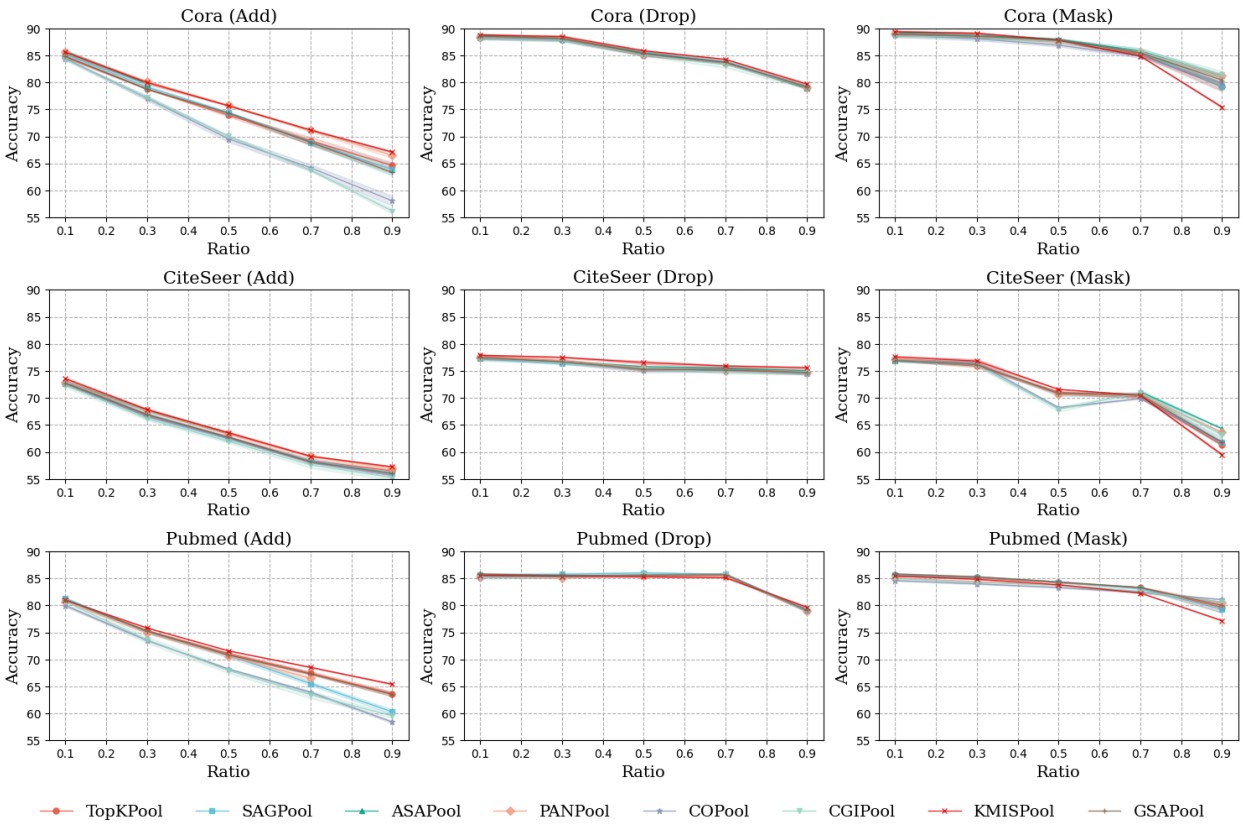

Figure 3: Performance of different approaches w.r.t. different rates of random noise.

Table 5: Results of **graph classification under random noise attack** for different pooling methods.

| Dataset | Ptb Method | TopKPool | SAGPool | ASAPool | PANPool | KMISPool | DiffPool | MincutPool | JustBalancePool |
|---|---|---|---|---|---|---|---|---|---|
| PROTEINS | ADD | 73.58±5.77 | 72.76±3.87 | 74.59±3.74 | 39.63±42.27 | 71.14±0.57 | 73.78±2.59 | 72.36±2.92 | **75.00±1.49** |
| | DROP | 71.95±2.28 | **73.58±2.55** | 72.76±1.88 | 38.62±42.97 | 71.34±1.32 | 71.34±1.80 | 71.75±2.07 | 73.37±1.25 |
| | MASK | 72.56±3.11 | **73.78±3.45** | 71.14±2.55 | 72.88±4.71 | 72.97±2.24 | 72.97±0.29 | 72.36±3.24 | 70.12±2.28 |
| NCI1 | ADD | 65.91±0.65 | 67.53±2.25 | 71.42±1.53 | 66.29±0.40 | 72.66±1.40 | 72.66±0.20 | 70.77±1.80 | **73.96±0.08** |
| | DROP | 63.16±1.52 | 61.64±1.46 | 64.07±0.73 | 65.53±1.31 | **73.58±3.00** | 66.18±2.22 | 65.05±0.73 | 64.18±1.61 |
| | MASK | 63.86±1.39 | 63.16±2.10 | 66.94±0.87 | 66.18±0.61 | 65.15±2.30 | **68.23±2.73** | 67.10±2.07 | 67.91±1.98 |
| NCI109 | ADD | 66.18±0.73 | 68.55±1.13 | 69.62±2.11 | 64.84±2.74 | **75.32±0.99** | 73.33±0.97 | 71.34±2.89 | 71.29±2.79 |
| | DROP | 63.59±1.40 | 64.61±1.49 | 64.24±1.86 | 65.64±1.26 | **73.85±2.95** | 66.18±2.22 | 65.05±0.73 | 64.18±1.61 |
| | MASK | 65.22±1.97 | 66.61±0.92 | 65.65±1.08 | 66.29±0.91 | 66.34±1.85 | 66.99±2.71 | **68.23±0.47** | **66.61±3.43** |
| MUTAG | ADD | **86.21±5.63** | 79.31±2.82 | 75.86±10.15 | 68.97±4.88 | 80.46±4.30 | 72.41±9.75 | 78.16±4.30 | 68.97±4.88 |
| | DROP | **87.36±4.30** | 63.22±13.9 | 72.41±14.90 | 68.97±2.82 | 80.46±4.30 | 78.16±1.63 | 74.71±9.89 | 75.86±11.26 |
| | MASK | **78.16±16.26** | 64.37±9.05 | 60.92±7.09 | 71.26±3.25 | 83.91±8.13 | 78.16±4.30 | 77.01±3.25 | 70.11±4.30 |

compared to node dropping pooling. *Secondly*, among node dropping pooling methods, KMISPool generally performs the best. However, for small datasets such as MUTAG, TopKPool achieves the highest performance under noise attacks, because its node selection mechanism is less sensitive to local noise variations (Knyazev et al., 2019). *Thirdly*, noise attacking increases the performance fluctuations of pooling methods, making their prediction results more unstable. *Fourthly*, in larger datasets such as PROTEINS, NCI1, and NIC109, dropping edges has a greater impact on performance, whereas for MUTAG, masking node features has a more significant effect.

Table 6 presents the results of the robustness analysis for node-level tasks. From Table 6, we observe the following: *Firstly*, random attacks on the graph lead to a decrease in performance on node classification tasks, with different types of attacks causing varying degrees of performance degradation. Randomly adding edges has the most negative impact on performance, while randomly deleting edges has the least impact.

Table 6: Results of **node classification under random noise attack** for different pooling methods.

| Dataset | Ptb Method | TopKPool | SAGPool | ASAPool | PANPool | COPool | CGIPool | KMISPool | GSAPool | HGPSLPool |
|---------|-----------|----------|---------|---------|---------|--------|---------|----------|---------|-----------|
| Cora | ADD | 73.90±0.24 | 74.41±0.12 | OOM | **75.75±0.14** | 69.52±0.58 | 70.01±0.24 | 75.64±0.03 | 74.29±0.15 | 75.58±0.14 |
| | DROP | 85.01±0.09 | 85.45±0.36 | 85.40±0.19 | 85.11±0.20 | 85.06±0.13 | 85.04±0.23 | **85.83±0.21** | 85.30±0.33 | 85.60±0.19 |
| | MASK | 87.70±0.16 | 87.75±0.18 | **87.94±0.12** | 87.48±0.24 | 86.88±0.26 | 87.42±0.03 | 87.81±0.17 | 87.83±0.37 | 87.59±0.24 |
| Citeseer | ADD | 62.64±0.19 | 62.47±0.41 | 63.43±0.14 | 63.38±0.37 | 62.62±0.21 | 61.94±0.29 | **63.52±0.20** | 62.69±0.23 | 63.42±0.21 |
| | DROP | 75.31±0.26 | 75.50±0.19 | 75.81±0.04 | 75.52±0.10 | 75.18±0.46 | 75.34±0.12 | **76.54±0.32** | 75.32±0.29 | 76.00±0.21 |
| | MASK | 73.29±0.27 | 73.41±0.25 | 73.57±0.17 | 73.42±0.20 | 73.54±0.44 | 73.28±0.49 | **73.63±0.10** | 73.45±0.20 | 73.30±0.24 |
| Pubmed | ADD | 71.06±0.25 | 70.75±0.41 | OOM | 70.62±0.12 | 68.21±0.11 | 67.92±0.45 | **71.59±0.01** | 70.83±0.17 | OOM |
| | DROP | 85.46±0.09 | **86.03±0.12** | OOM | 85.55±0.04 | 85.68±0.04 | 85.59±0.13 | 85.30±0.06 | 85.59±0.06 | OOM |
| | MASK | 84.24±0.04 | 84.34±0.06 | OOM | 83.75±0.06 | 83.31±0.07 | 83.78±0.17 | 83.83±0.02 | **84.36±0.11** | OOM |

Table 7: Results of **graph classification under distribution shifts**. Size and density denote two types of shifts across training and test datasets. Micro-F1 and Macro-F1 metrics are provided for each shift type.

| Method | D&D | | | | NCI1 | | | |
|--------|-----|-----|-----|-----|------|-----|-----|-----|
| | Size | | Density | | Size | | Density | |
| | Micro-F1 | Macro-F1 | Micro-F1 | Macro-F1 | Micro-F1 | Macro-F1 | Micro-F1 | Macro-F1 |
| *Node Drop Pooling* | | | | | | | | |
| TopKPool | 68.08±1.60 | 63.69±1.13 | 31.98±15.37 | 29.43±15.16 | 25.89±0.46 | 24.98±0.43 | 53.48±2.11 | 51.02±3.14 |
| SAGPool | 81.36±2.49 | 74.47±1.48 | 55.37±0.89 | 50.14±0.97 | 25.08±2.39 | 23.90±2.95 | 47.00±3.60 | 45.86±3.49 |
| ASAPool | OOT | OOT | OOT | OOT | 26.29±3.66 | 25.29±4.42 | 53.17±1.85 | 51.34±1.18 |
| PANPool | 77.68±8.37 | 71.44±6.50 | 41.92±10.72 | 40.36±10.05 | 25.00±0.00 | 23.74±0.08 | 52.08±2.24 | 53.09±2.35 |
| COPool | 64.41±5.66 | 60.37±3.74 | 47.91±2.51 | 44.30±1.81 | 27.99±3.86 | 27.17±4.48 | 54.67±2.22 | 53.09±2.35 |
| CGIPool | 75.99±6.65 | 69.62±5.58 | 56.38±1.76 | 51.10±1.45 | 28.16±5.18 | 27.26±5.81 | 56.20±0.86 | 53.93±1.12 |
| KMISPool | 80.23±5.24 | 73.30±4.04 | 54.58±5.26 | 49.81±3.62 | 50.97±9.51 | 49.02±7.87 | 55.42±1.47 | 51.27±0.30 |
| GSAPool | 58.19±26.99 | 53.06±25.69 | 33.79±20.93 | 29.39±19.01 | 26.21±1.24 | 25.19±1.56 | 50.31±3.39 | 49.32±2.65 |
| HGPSLPool | **85.59±1.20** | **78.34±1.66** | 52.43±2.25 | 49.59±1.89 | 19.66±0.52 | 17.52±0.66 | 56.95±1.64 | 51.93±1.14 |
| *Node Clustering Pooling* | | | | | | | | |
| AsymCheegerCutPool | 74.47±0.06 | 73.60±0.06 | 86.13±0.00 | 50.86±0.01 | 48.87±3.06 | 45.42±1.65 | 70.01±0.00 | 46.95±0.00 |
| DiffPool | 73.87±0.02 | 73.35±0.02 | 86.49±0.02 | 47.44±0.01 | 19.50±0.00 | 16.63±0.01 | 69.53±0.00 | 50.87±0.15 |
| MincutPool | 69.97±0.17 | 67.95±0.17 | **87.39±0.00** | 46.63±0.00 | 19.58±0.00 | 16.69±0.00 | 68.64±0.00 | 50.31±0.09 |
| DMoNPool | 72.67±0.11 | 72.25±0.13 | 82.52±0.00 | **54.21±0.00** | 79.29±0.04 | **64.30±0.00** | 68.92±0.00 | 48.76±0.02 |
| HoscPool | 70.27±0.01 | 69.20±0.00 | **87.39±0.00** | 46.63±0.00 | 24.60±0.27 | 23.39±0.36 | **70.48±0.01** | **56.55±0.35** |
| JustBalancePool | 68.77±0.00 | 67.82±0.02 | **87.39±0.00** | 46.63±0.00 | 19.98±0.00 | 17.28±0.01 | 68.64±0.00 | 50.31±0.09 |
| ParsPool | 68.36±1.60 | 62.50±1.76 | 63.06±4.84 | 48.35±0.86 | **52.59±4.46** | 49.95±3.32 | 56.27±1.97 | 52.96±0.88 |

*Secondly*, for larger graphs such as Cora, Citeseer, and Pubmed, KMISPool performs the best, whereas for smaller graphs such as Cornell, Texas, and Wisconsin, ASAPool performs better. *Thirdly, for graphs with a large diameter, such as PubMed (Diameter=18), SAGPool performs best when edges are deleted or node features are masked. SAGPool leverages graph convolution to compute self-attention scores, enabling node importance to depend on both its intrinsic features and its topological relationships with other nodes. This mechanism allows SAGPool to effectively capture long-range dependencies in large-diameter graphs, as the attention mechanism can focus on important nodes even when long connection paths exist.* Appendix C.1 provides results for the robustness analysis of node-level tasks.

As shown in Figure 3, the model's performance generally declines as the noise intensity increases. It is observed that, at the same level of noise, the impact on accuracy is more pronounced on smaller datasets Cora and CiteSeer, while it is relatively minor on larger dataset Pubmed. Among the three types of noise, although the accuracy of nearly all methods decreases amidst fluctuations, KMISPool and PANPool exhibit the strongest robustness, while COPool performs relatively poorly, despite the fact that most pooling methods show very similar performance.

## 4.4 Generalizability Analysis

Table 7 and Table 8 presents the performance of different graph pooling methods under out-of-distribution shifts. For the graph-level datasets D&D and NCI1, we implement two types of distribution shifts. The first type is based on the number of nodes, where the smallest 50% of graphs by node count are used as the training set, and the largest 20% as the test set, with the remainder serving as the validation set (Bevilacqua et al., 2021; Chen et al., 2022). Following the same criteria, the second type of out-of-distribution shifts are generated based on graph density (Chen et al., 2022). For the node-level datasets Cora and Citeseer, the

Table 8: Results of **node classification under distribution shifts**. Degree and closeness denote two types of shifts across training and test datasets.

| Method | Cora | | | | Citeseer | | | |
| --- | --- | --- | --- | --- | --- | --- | --- | --- |
| | Degree | | Closeness | | Degree | | Closeness | |
| | Micro-F1 | Macro-F1 | Micro-F1 | Macro-F1 | Micro-F1 | Macro-F1 | Micro-F1 | Macro-F1 |
| TopKPool | 83.65±0.50 | 82.29±0.47 | 83.21±0.18 | 81.98±0.31 | 67.27±0.35 | 63.60±0.26 | 72.28±0.67 | 65.64±1.00 |
| SAGPool | 84.19±0.12 | 82.73±0.27 | 81.68±1.54 | 80.34±1.55 | 65.46±0.61 | 62.22±0.49 | 72.28±1.10 | 66.35±1.87 |
| ASAPool | 83.16±0.00 | 82.24±0.13 | 84.10±0.37 | 82.97±0.32 | 67.47±0.23 | 63.94±0.31 | 72.84±0.49 | 65.27±1.74 |
| PANPool | 84.00±0.37 | 82.83±0.37 | **84.44±0.30** | **83.44±0.36** | 67.63±0.26 | 63.85±0.28 | **73.45±0.86** | 67.48±1.75 |
| COPool | 83.90±0.12 | 82.62±0.13 | 81.14±1.72 | 80.00±1.01 | 66.43±0.59 | 62.93±0.42 | 72.80±0.00 | 67.14±0.42 |
| CGIPool | 83.21±0.77 | 82.24±0.82 | 82.27±0.55 | 80.91±0.80 | 65.66±1.34 | 62.14±1.46 | 72.36±0.46 | 66.80±0.91 |
| KMISPool | 84.00±0.18 | 82.57±0.15 | 83.55±0.39 | 82.36±0.41 | 67.35±0.15 | 63.69±0.14 | 72.72±0.30 | 66.51±0.39 |
| GSAPool | 83.70±0.07 | 82.19±0.27 | 83.16±0.32 | 81.89±0.37 | 67.07±0.49 | 63.56±0.42 | 72.84±0.44 | 66.33±0.37 |
| HGPSLPool | **84.19±0.00** | **82.83±0.04** | 83.46±0.79 | 82.32±0.89 | **67.67±0.15** | **64.13±0.16** | 73.20±0.46 | **67.95±0.40** |

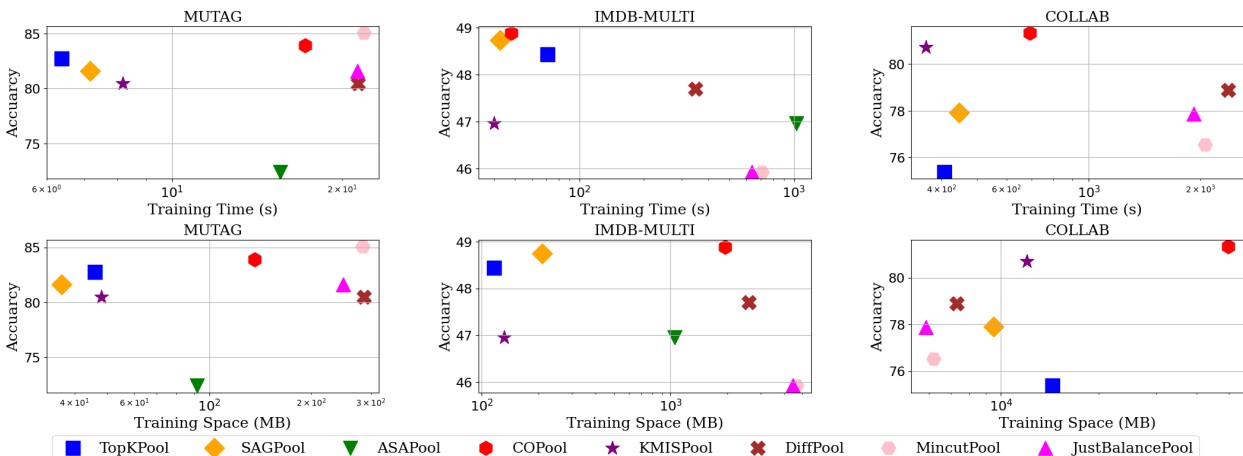

Figure 4: Comparison of performance, training time, and memory usage across different approaches.

first type of out-of-distribution shift is the top 50% of nodes with the highest degrees as the training set, the bottom 25% with the lowest degrees as the test set, and the remaining nodes as the validation set. The second type is based on closeness centrality (the reciprocal of the sum of the shortest path lengths from a node to all other nodes). We use the 50% of nodes with the lowest closeness as the training set, the 25% with the highest closeness as the test set, and the remaining nodes as the validation set. For further details and more experiments for the generalizability analysis, please refer to the Appendix A.4 and Appendix C.2.

From Tables 7 and 8, we have the following observations. *Firstly*, node-level out-of-distribution shifts also reduce the performance of pooling models, but the extent of this reduction is smaller compared to out-of-distribution shifts in graph classification tasks. The potential reason is that, in node-level tasks, the propagation of information are usually confined to the local neighborhood of nodes, whereas graph-level tasks require handling information spread over a larger scope. *Secondly*, Macro-F1 is generally lower than Micro-F1, which indicates that the model has weaker recognition capabilities for minority classes. *Thirdly*, node clustering pooling exhibits better generalizability than node dropping pooling in graph classification tasks. *Fourthly*, HGPSLPool and PANPool exhibit the best performance, potentially due to the fact that HGPSLPool combines graph convolution with spectral clustering, enabling it to better capture higher-order relationships and local topological structures, which is advantageous in node-level tasks. Meanwhile, PAN-Pool utilizes an adaptive pooling strategy that adjusts the pooling method to suit different node feature distributions, enhancing the model's robustness and generalization capability under out-of-distribution conditions.

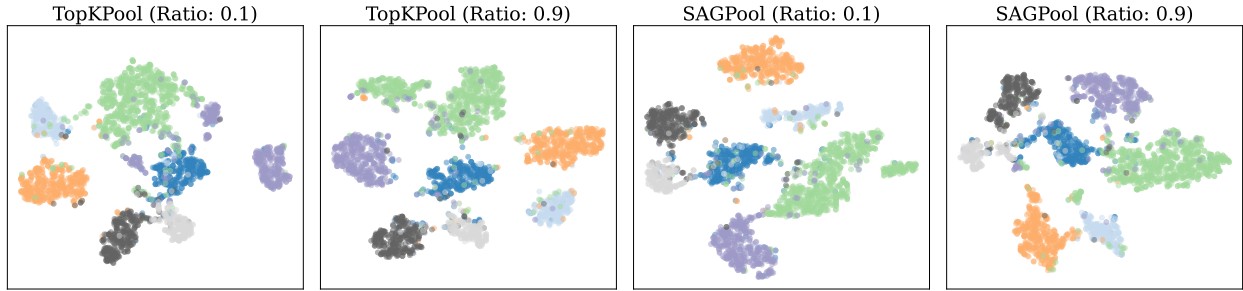

Figure 5: The t-SNE visualization w.r.t. different pooling ratios of TopKPool and SAGPool.

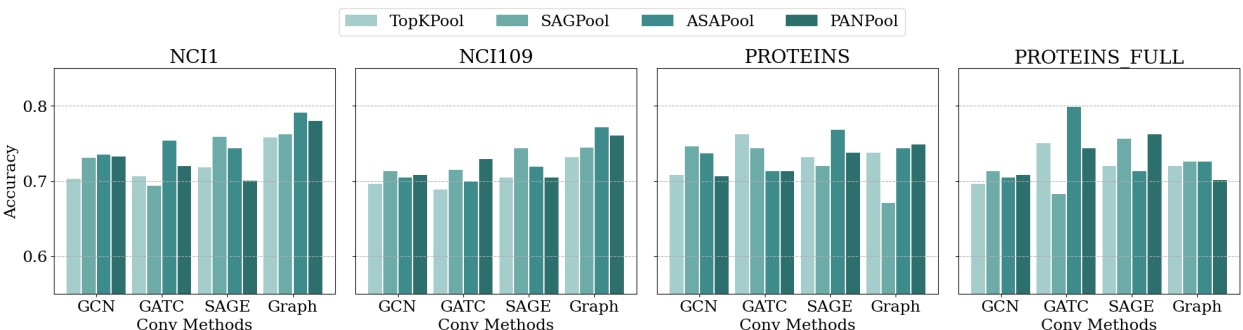

Figure 6: Performance w.r.t. graph convolution backbones for different pooling methods.

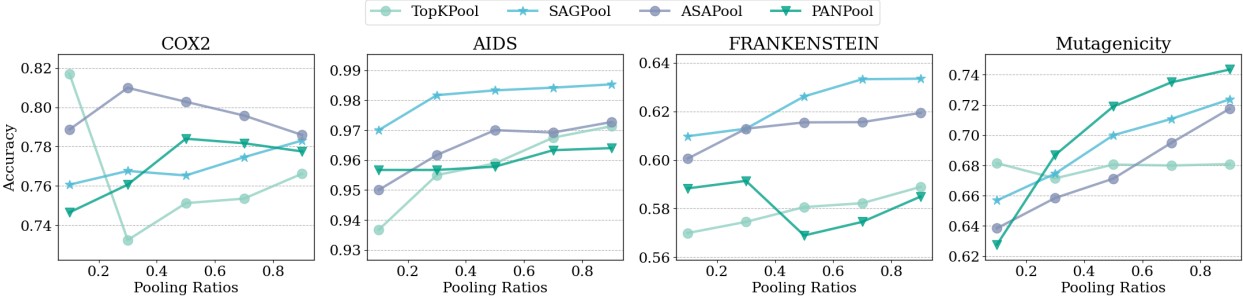

Figure 7: Performance w.r.t. different pooling ratios for four pooling methods.

### 4.5 Further Analysis

**Efficiency Comparison.** In this part, we conduct an efficiency analysis of graph pooling methods on the MUTAG, IMDB-MULTI, and COLLAB datasets. We calculate the time of the algorithms by measuring the duration needed to complete 200 epochs of training with the 512 batch size. For space efficiency, we compute the GPU memory utilization during the training process. From Figure 4, it can be observed that ASAPool, DiffPool, MincutPool, and JustBalancePool exhibit significantly higher time and space costs. In contrast, node dropping pooling methods such as TopKPool, SAGPool, and KMISPool demonstrate lower time and space costs. The underlying reason is that node clustering pooling methods require converting graph data into an adjacency matrix form and simplifying the graph through clustering rather than directly removing nodes.

**Visualization.** Figure 5 shows the t-SNE visualization for TopKPool and SAGPool under different pooling ratios. From the results, we observe that as the pooling ratio increases from 0.1 to 0.9, the different classes form more distinct clusters in the t-SNE plot when the pooling ratio is low. As the pooling ratio increases,

the model retains more nodes, leading to a greater overlap between nodes of different classes and a reduction in inter-class separability. Moreover, when the pooling ratio is 0.9, SAGPool shows a higher degree of class separability compared to TopKPool.

**Backbone Analysis.** Figure 6 presents the performance of four pooling methods based on GCNConv (Kipf & Welling, 2016), GATConv (Veličković et al., 2017), SAGEConv (Hamilton et al., 2017), and Graph-Conv (Morris et al., 2019) on four datasets, NCI1, NCI109, PROTEINS, and PROTEINS_FULL. On average, as the backbone models change, most pooling methods exhibit significant performance fluctuations, and no single backbone model consistently maintains a leading position. Except for the PROTEINS_FULL, the performance of GraphConv is relatively better.

**Parameter Analysis.** Figure 7 shows the performance of four pooling methods on the COX2, AIDS, FRANKENSTEIN, and Mutagenicity datasets. From the results, we observe that as the pooling rate increases from 0.1 to 0.9, the performance increases before reaching saturation in most cases. The performance variation among different pooling methods is significant as the pooling ratio changes, it is necessary to adjust the pooling ratio when employing pooling methods.

## 5 Related Work

### 5.1 Graph Classification and Graph Regression

Graphs provide an effective tool to represent interaction among different objects (Wu et al., 2020b). Graph classification (Baek et al., 2021) is a fundamental graph machine learning problem, which aims to classify each graph into its corresponding category. The majority of current works adopt message passing mechanisms (Wu et al., 2020b), where each node receives information from its neighbors in a recursive manner. Then, a graph readout function is adopted to summarize all node representations into a graph-level representation for downstream classification. Graph classification has extensive applications in various domains such as molecular property prediction (Wieder et al., 2020) and protein function analysis (Mills et al., 2018). Graph regression (Qin et al., 2023) is close to graph classification which maps graph-level data into continuous vectors. Researchers usually utilize graph regression to formulate molecular property predictions (Mqawass & Popov, 2024). Graph pooling has been an important topic in graph-level tasks (Knyazev et al., 2019; Lee et al., 2019; Ranjan et al., 2020; Ma et al., 2020b; Zhou et al., 2022; Pang et al., 2021; Bacciu et al., 2023; Zhang et al., 2020a; 2019; Hansen & Bianchi, 2023; Ying et al., 2018a; Bianchi et al., 2020; Tsitsulin et al., 2023a; Duval & Malliaros, 2022; Bianchi, 2022), which generally utilize a hierarchical way to refine the graph structures (Bianchi & Lachi, 2024; Liu et al., 2022b). In this work, we generally study the performance of graph pooling on graph-level tasks and validate the effectiveness of graph pooling approaches in most cases.

### 5.2 Node Classification

Node classification aims to classify each node in a graph based on its attributes and relationship with the other nodes (Xiao et al., 2022; Ju et al., 2024; Zhong et al., 2022; Prieto et al., 2023). Node classification has various applications in the real world, including social network analysis (Camacho et al., 2020), knowledge graphs (Ye et al., 2022), bioinformatics (Bhagat et al., 2011) and online commerce services (Yu et al., 2023). Graph neural networks have been widely utilized to solve the problem by learning semantics information across nodes by neighborhood propagation. Since graph pooling would reduce the number of nodes, recent works utilize a U-Net architecture (Gao & Ji, 2019), which involves down-sampling and up-sampling with residue connections. In this work, we systematically evaluate the performance of graph pooling on node classification and observe that graph pooling has limited improvement compared with basic graph neural network architectures.

### 5.3 Previous Benchmark Research

Previous studies have built benchmarks for graph-related tasks (Errica et al., 2019; Hu et al., 2020b; Tönshoff et al., 2023). In particular, Errica et al. (2019) is a benchmark including six different GNN models across nine commonly used TUDataset datasets. Open Graph Benchmark (OGB) (Hu et al., 2020b) evaluates

different graph neural network approaches experiments on graph classification, graph regression, and node classification. Errica et al. (2019) only involve one graph pooling method and Hu et al. (2020b) does not involve any graph pooling methods. In comparison, our method focuses on graph pooling techniques rather than graph neural networks. Moreover, our benchmark explores the robustness of these methods by introducing noise attacks in both graph classification and node classification tasks and investigating their generalizability through out-of-distribution shifts.

# 6 Summary of Observations and Guidance

Overall, we summarize the main insights basde on our experimental results. Firstly, for graph classification and regression tasks on small-scale graphs with fewer nodes and edges, we recommend using feature extractors that aggregate information from neighboring nodes, such as GCNConv, along with MLPs, to capture both the topological semantics of the graph and the feature semantics of the nodes (e.g., GSAPool and CGIPool). This is because, in small-scale graphs, the topological information of the nodes is relatively easy to obtain. Conversely, for graph classification and regression tasks on large-scale graphs with a substantial number of nodes and edges, we suggest focusing on both local structures and global connectivity patterns (e.g., ParsPool and SEPool), paying attention not only to the topological information of the nodes but also to the link structures of the edges.

Secondly, for node classification tasks on graphs with strong connectivity, we recommend designing pooling structures that aggregate feature information from nodes and their neighbors during the pooling process (e.g., GSAPool and SAGPool). The reason is that in such graphs, nodes are closely connected, and the information between neighboring nodes is richer and easier to extract. On the other hand, for node classification tasks on graphs with weak connectivity, we suggest designing adaptive pooling algorithms that focus more on global information and select nodes more uniformly across the graph (e.g., KMISPool). This is because weaker connectivity implies a more dispersed graph topology, and pooling methods that focus solely on local information may lose representativeness in node selection.

Thirdly, for node classification tasks on graphs with a large diameter, we recommend combining attention mechanisms with algorithms that uniformly select nodes (e.g., SAGPool and KMISPool). The attention mechanism can capture important nodes in graphs with long-range topological dependencies, while uniform node selection allows for the capture of long-distance dependencies and enhances the representativeness of the selected nodes.

Fourthly, when facing graph structural noise attacks, for node classification tasks, we recommend designing pooling algorithms that uniformly sample nodes from the graph. This approach is less sensitive to graph edges and node features compared to algorithms that aggregate local information (e.g., KMISPool). For graph classification tasks, we suggest aggregating neighbor features when designing node dropping pooling methods (e.g., DiffPool, MincutPool, and JustBalancePool). This dilutes the outliers of individual nodes, making the representations of higher-level supernodes more dependent on the overall neighborhood information. When designing node clustering pooling, multiple rounds of neighbor message passing should be performed, as noise features are averaged or weighted multiple times during propagation, enhancing noise suppression. Additionally, when designing the loss function, attention should be paid to balancing the size of each cluster to prevent noise nodes or edges from causing the model to collapse into extreme allocations.

Fifthly, on large-scale graphs, graph pooling can improve computational efficiency and allow graph neural networks to capture more important nodes. Although node clustering pooling shows strong performance (e.g., AsymCheegerCutPool, DiffPool, MincutPool, ParsPool), its computational and spatial costs are higher than those of node dropping pooling (e.g., TopKPool, SAGPool). The reasoning is that the pooling structure requires a soft assignment matrix to determine the clustering relationships of all nodes. Future research could focus on how to efficiently compute this soft assignment matrix.

# 7 Conclusion

In this paper, we construct the first graph pooling benchmark that includes 17 state-of-the-art approaches and 28 different graph datasets across graph classification, graph regression, and node classification. We find that node clustering pooling methods outperform node dropping pooling methods in terms of robustness and generalizability, but at the cost of higher computational expenses. This benchmark systematically analyzes the effectiveness, robustness, and generalizability of graph pooling methods. We also make our benchmark publically available to advance the fields of graph machine learning and applications. One limitation of our benchmark is the lack of more complicated settings under label scarcity. In future works, we would extend our graph pooling benchmark to more realistic settings such as semi-supervised learning and few-shot learning.

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

# Appendix

## A    Detailed Description of Datasets

### A.1    Graph Classification

Table A.1 provides descriptive statistics of the selected datasets, revealing that our chosen datasets encompass graph data of varying scales and features. This diversity establishes a robust foundation for benchmarking. The following are detailed descriptions of these datasets:

**Ogbg-molpcba** comprises a collection of 437,929 molecules, each represented as a graph where nodes are atoms and edges indicate chemical bonds between atoms. Each node is associated with features such as atom type, valence, and charge. The dataset involves 128 biological activity labels, each representing a binary classification task that indicates whether a molecule exhibits a specific biological activity (Hu et al., 2020b).

**PROTEINS** represents protein structures; nodes denote secondary structure elements (SSEs) and the edges indicate the relationships between these SSEs that are in close proximity. The primary goal of this dataset is to assist in the classification of proteins into different structural classes based on their amino acid sequences and structure— structural characteristics. Each graph's label is the protein class, so the dataset covers diverse protein structures (Borgwardt et al., 2005).

**PROTEINS_FULL** is an extended version of PROTEINS. Each graph directly represents a protein structure: nodes correspond to SSEs like alpha helices and beta sheets (Borgwardt et al., 2005).

**NCI1** is a collection of chemical compound graphs. Originating from the National Cancer Institute (NCI) database, each graph sample is a compound in which nodes represent atoms and edges represent the bonds between them. The dataset is binary-class labeled, indicating biological activity via compounds' anti-cancer activity against specific cell lines (Wale et al., 2008).

**NCI109** is also a collection of chemical compound graphs derived from NCI. Similarly, each node in the graph denotes an atom and each edge denotes a bond. The two classes in NCI109 are about compounds' ability to inhibit or interact with the specified cancer cell line (Wale et al., 2008).

**MUTAG** consists of 188 chemical molecule graphs, where each node represents an atom. The nodes have different atomic types, such as carbon, nitrogen, oxygen, etc. Edges represent chemical bonds between atoms, such as single or double bonds, indicating their connections in the molecule. The objective is to predict whether each molecule is mutagenic, with positive labels indicating mutagenic molecules and negative labels indicating non-mutagenic molecules (Morris et al., 2020).

**D&D** is a dataset of protein structure graphs for graph classification. Each graph in this dataset represents a protein, with nodes corresponding to amino acids and edges corresponding to the spatial or sequential proximity between these amino acids. The primary objective of the D&D dataset is to classify proteins into one of two categories: enzymes or non-enzymes (Shervashidze et al., 2011).

**IMDB-B** is a collection of social network graphs derived from the Internet Movie Database (IMDB). Each graph is about a collaboration network from movies whereby nodes stand for actors or actresses and edges indicate that the two actors appeared in the same movie — this dataset comprises two classes reflecting the movie genres (Cai & Wang, 2018).

**IMDB-M**, similar to IMDB-B, represents each movie as a graph where the nodes represent actors and the edges represent co-appearances of actors in the same movie. However, the nodes in IMDB-M are categorized into three classes, and it includes a larger number of actors (Cai & Wang, 2018).

**COLLAB** consists of 5,000 graphs, each representing a collaboration network of a group of authors in different research fields. In each graph, nodes represent authors, and edges represent collaborations between authors, indicating that the connected authors have co-authored at least one paper. The graphs have three classes, each corresponding to an academic research field. (Morris et al., 2020).

Table A.1: Summary statistics of datasets for **graph classification**. CC denotes the clustering coefficient, and Diameter representing the maximum value of the shortest path between any two nodes in the graph.

| Datasets | Graphs | Classes | Avg. Nodes | Avg. Edges | Node Attr. | Avg. Diameter | Avg. Degree | Avg. CC |
|---|---|---|---|---|---|---|---|---|
| Ogbg-molpcba | 437,929 | 2*128 | 26.00 | 27.50 | - | 12.00 | 2.20 | 0.00 |
| PROTEINS | 1,113 | 2 | 39.06 | 72.82 | + (1) | 11.14 | 3.73 | 0.51 |
| PROTEINS_full | 1,113 | 2 | 39.06 | 72.82 | + (29) | 11.14 | 3.73 | 0.51 |
| NCI1 | 4,110 | 2 | 29.87 | 32.30 | - | 11.45 | 2.16 | 0.00 |
| NCI109 | 4,127 | 2 | 29.68 | 32.13 | - | 11.21 | 2.16 | 0.00 |
| MUTAG | 188 | 2 | 17.90 | 39.60 | + (7) | 8.22 | 2.19 | 0.00 |
| D&D | 1,178 | 2 | 284.32 | 715.66 | - | 16.45 | 4.92 | 0.48 |
| IMDB-B | 1,000 | 2 | 19.77 | 96.53 | - | 1.86 | 8.89 | 0.95 |
| IMDB-M | 1,500 | 3 | 13.00 | 65.94 | - | 1.47 | 8.10 | 0.97 |
| COLLAB | 5,000 | 3 | 74.49 | 2457.22 | - | 1.86 | 37.36 | 0.89 |
| COX2 | 467 | 2 | 41.22 | 43.45 | + (3) | 13.79 | 2.11 | 0.00 |
| AIDS | 2,000 | 2 | 15.69 | 16.20 | + (4) | 6.56 | 2.01 | 0.01 |
| FRANKENSTEIN | 4,337 | 2 | 16.90 | 17.88 | + (780) | 7.86 | 2.06 | 0.01 |
| Mutagenicity | 4,337 | 2 | 30.32 | 30.77 | - | 9.10 | 2.04 | 0.00 |

**COX2** consists of 467 graphs, where each graph corresponds to a molecule. The nodes represent atoms, and the edges represent chemical bonds, and the graph label indicates whether the molecule is a COX-2 inhibitor (Sutherland et al., 2003).

**AIDS** consists of 2,000 graphs. Each graph corresponds to a molecule, where the nodes represent individual atoms and the edges represent chemical bonds between these atoms. Here, we want to predict the inhibitory effect of molecules on HIV based on their structure. (Riesen & Bunke, 2008).

**FRANKENSTEIN** consists of 4,337 graphs. Each graph in this dataset represents a chemical compound, where the nodes correspond to atoms, and the edges represent the bonds between them. The graph labels indicate whether the molecule is considered an active compound (Orsini et al., 2015).

**Mutagenicity** contains 4,337 molecular graphs. In Mutagenicity, each graph represents a molecule, where the nodes are atoms and the edges denote chemical bonds between the nodes. The classification goal is to predict whether a molecule graph is mutagenic or not (Debnath et al., 1991).

### A.2   Graph Regression

Table A.2 provides an overview of the selected datasets. A more detailed description is provided below.

**QM7 and QM8** are benchmark datasets in computational chemistry, designed to facilitate the development and evaluation of machine learning approaches for quantum mechanical property prediction. It contains approximately 7,165 (QM7) and 21,786 (QM8) molecular structures, each characterized by their calculated properties using quantum chemistry methods, specifically focusing on electronic spectra (Wu et al., 2018; Montavon et al., 2013).

**BACE** is a collection of biochemical data used to evaluate computational methods for drug discovery. The dataset includes a total of 1,522 compounds, each annotated with their binding affinities, as well as molecular descriptors and fingerprints to facilitate the development and assessment of machine learning modelsa (Wu et al., 2018; Ciordia et al., 2016).

**ESOL** is a prominent resource in cheminformatics, designed for evaluating machine learning models on the prediction of aqueous solubility of small molecules. The dataset, derived from the work of Delaney, encompasses a diverse range of chemical compounds with experimentally determined solubility values expressed in logS, where S is the solubility in mols per liter. It includes 1128 compounds, serving as a benchmark for solubility prediction tasks (Delaney, 2004; Wu et al., 2018).

**FreeSolv** is a dataset containing hydration-free energies for small molecules in an aqueous solution. It comprises data for a wide range of organic molecules, providing both experimental values and calculated predictions based on molecular simulations (Mobley & Guthrie, 2014; Wu et al., 2018).

Table A.2: Details of datasets for **graph regression**.

| Datasets | Tasks | Compounds | Split | Avg. Nodes | Avg. Edges | Avg. Diameter | Avg. Degree | Avg. CC |
|---|---|---|---|---|---|---|---|---|
| QM7 | 1 | 7,165 | Scaffold | 6.79 | 6.44 | 4.21 | 1.89 | 0.06 |
| QM8 | 12 | 21,786 | Random | 7.77 | 8.09 | 4.35 | 2.08 | 0.09 |
| BACE | 1 | 1,522 | Scaffold | 34.09 | 36.86 | 4.35 | 2.08 | 0.01 |
| ESOL | 1 | 1,128 | Random | 13.30 | 13.69 | 7.02 | 1.98 | 0.00 |
| FreeSolv | 1 | 643 | Random | 8.76 | 8.43 | 5.06 | 1.84 | 0.00 |
| Lipophilicity | 1 | 4,200 | Random | 27.04 | 29.50 | 13.85 | 2.18 | 0.00 |

Table A.3: Summary statistics of datasets for **node classification**.

| Datasets | Number of Nodes | Number of Edges | Number of Features | Number of Classes | Diameter | Avg. Degree | Avg. CC |
|---|---|---|---|---|---|---|---|
| Ogbn-arxiv | 169,343 | 1,166,243 | 128 | 40 | 23 | 13.72 | 0.23 |
| Cora | 2,708 | 10,556 | 1,433 | 7 | NA | 3.90 | 0.24 |
| CiteSeer | 3,327 | 9,104 | 3,703 | 6 | NA | 2.74 | 0.14 |
| PubMed | 19,717 | 88,648 | 500 | 3 | 18 | 4.50 | 0.06 |
| Cornell | 183 | 298 | 1,703 | 5 | 8 | 3.06 | 0.17 |
| Texas | 183 | 325 | 1,703 | 5 | 8 | 3.22 | 0.20 |
| Wisconsin | 251 | 515 | 1,703 | 5 | 8 | 3.71 | 0.21 |
| Github | 37,700 | 578,006 | 0 | 2 | 7 | 15.33 | 0.01 |

**Lipophilicity** is primarily utilized for studying and evaluating molecular lipophilicity. This dataset comprises 4,200 compounds sourced from the ChEMBL database, with experimentally measured partition coefficient (logD) values that reflect the distribution behavior of compounds in a water-octanol system (Lukashina et al., 2020; Wu et al., 2018).

### A.3 Node Classification

Table A.3 presents descriptive statistics of the seven datasets used for node classification. It is evident that there is a significant variance in the scale of the selected datasets, each possessing distinct characteristics. Further background information and details about these datasets are provided below.

**Ogbn-arxiv** comprises a collection of 169,343 scientific publications classified into 40 distinct categories. Each paper is represented by a node with a 128-dimensional feature, which comes from the average of word embeddings in the corresponding title and abstract. Edges indicate citation relationships between papers (Hu et al., 2020b).

**Cora** comprises a collection of 2,708 scientific publications classified into seven distinct categories. Each publication in the dataset is represented as a node in a citation network, where edges indicate citation relationships between papers (Yang et al., 2016).

**CiteSeer** is a widely used citation network dataset. It comprises scientific publications categorized into six classes, with each publication represented by a 3,327-dimensional binary vector recording the presence or absence of specific words (Yang et al., 2016).

**PubMed** consists of scientific publications from the PubMed database, categorized into three classes based on their Medical Subject Headings (MeSH) terms. Each node has a sparse bag-of-words vector derived from the content of the corresponding publication (Yang et al., 2016).

**Cornell, Texas, and Wisconsin** are made up of nodes that represent web pages and edges which denote hyperlinks between these pages. Each node has a class which denotes the topic of the web page; this allows tasks including node classification and link prediction to be performed. The datasets differ in size: Cornell and Texas each have 183 nodes while Wisconsin has 251 nodes (Pei et al., 2020).

**Github** includes node attributes representing the features of developers, such as their interests, skills, and contributions to various repositories. The edges within the network capture the interactions and collaborations of developers, creating a multi-faceted graph structure (Rozemberczki et al., 2021).

### A.4 Out-of-distribution shifts

**Size shifts.** For the selected datasets, including NCI1, D&D, NCI109, and IMDB-B, we utilized the data provided by the authors of size-invariant-GNNs (Bevilacqua et al., 2021). In this setup, the graphs with the smallest 50% of nodes are used as the training set, those with the largest 20% of nodes are used as the test set, and the remaining graphs were used as the validation set.

**Density shifts.** For the selected datasets, we divide the datasets based on graph density: the 50% of graphs with the lowest density are used as the training set, the 20% with the highest density are used as the test set, and the remaining graphs are used as the validation set. After applying density shifts, the following densities are observed: for D&D, the training set density is 0.0274, the validation set density is 0.0536, the test set density is 0.1142; for NCI1, the training set density is 0.1229, the validation set density is 0.1920, the test set density is 0.2786; for NCI109, the training set density is 0.1248, the validation set density is 0.1943, the test set density is 0.2770; for IMDB-B, the training set density is 0.6574, the validation set density is 1.1074, the test set density is 1.7427.

**Degree shifts.** For the selected datasets, we divide the datasets based on node degree: the 50% of nodes with the highest degree are used as the training set, the 25% with the lowest degree are used as the test set, and the remaining nodes are used as the validation set. After applying degree shifts, we can observe that: for Cora, the training set average degree is 5.9431, the validation set average degree is 2.4225, and the test set average degree is 1.2836; for Citeseer, the training set average degree is 4.3313, the validation set average degree is 1.3430, and the test set average degree is 0.9424; for Pubmed, the training set average degree is 7.9148, the validation set average degree is 1.1552, and the test set average degree is 1.0000; for Cornell, the training set average degree is 3.2198, the validation set average degree is 0.1111, and the test set average degree is 0.0000; for Texas, the training set average degree is 3.3626, the validation set average degree is 0.4222, and the test set average degree is 0.0000; for Wisconsin, the training set average degree is 3.7600, the validation set average degree is 0.7258, and the test set average degree is 0.0000.

**Closeness shifts.** For the selected datasets, we divide the datasets based on node closeness: the 50% of nodes with the lowest closeness are used as the training set, the 25% with the highest closeness are used as the test set, the remaining nodes used as the validation set. After applying closeness shifts, we can observe that: for Cora, the training set average closeness is 0.1076, the validation set average closeness is 0.1560, the test set average closeness is 0.1786; for Citeseer, the training set average closeness is 0.0150, the validation set average closeness is 0.0679, the test set average closeness is 0.0832; for Pubmed, the training set average closeness is 0.1448, the validation set average closeness is 0.1669, the test set average closeness is 0.1850; for Cornell, the training set average closeness is 0.2690, the validation set average closeness is 0.3754, the test set average closeness is 0.3896; for Texas, the training set average closeness is 0.2899, the validation set average closeness is 0.3887, the test set average closeness is 0.4047; for Wisconsin, the training set average closeness is 0.2630, the validation set average closeness is 0.3686, the test set average closeness is 0.3855.

## B  Additional Experimental Details

### B.1  Graph Classification

The classification model comprises three primary components: GCNConv layers, pooling methods, and a global average pooling layer. The hidden and output channels for this model are both set to 64. Initially, the data passes through three GCNConv layers with ReLU activation functions, followed by two pooling layers, before arriving at a global average pooling layer. The embedding output from this global layer undergoes further processing through a linear layer with ReLU activation, having dimensions (64, 32), followed by another linear layer without any activation function but with dimensions (32, number of classes). The final output can be available after applying softmax to the embedding output. All models use the Adam optimizer with a learning rate of 0.001 and are trained for 200 epochs by minimizing the negative log-likelihood loss function. For Ogbg-molpcba, the data is divided into training, validation, and test sets with an 80%, 10%, and 10% split, respectively. The remaining datasets are divided into training, validation, and test sets with a 70%, 15%, and 15% split. Each trial is repeated multiple times with different random seeds.

Table A.4: Details of hyperparameter tuning for different pooling methods

| Methods | | Hyperparameter space |
|---|---|---|
| *Node Dropping Pooling* | | |
| TopKPool | NIPS'19 | Pooling ratio: 0.1, 0.3, 0.5, 0.7, 0.9 |
| SAGPool | ICML'19 | Pooling ratio: 0.1, 0.3, 0.5, 0.7, 0.9 |
| ASAPool | AAAI'20 | Pooling ratio: 0.1, 0.3, 0.5, 0.7, 0.9 |
| PANPool | NIPS'20 | Pooling ratio: 0.1, 0.3, 0.5, 0.7, 0.9 |
| COPool | ECMLPKDD'22 | Pooling ratio: 0.1, 0.3, 0.5, 0.7, 0.9; K: 1, 2, 3 |
| CGIPool | SIGIR'22 | Pooling ratio: 0.1, 0.3, 0.5, 0.7, 0.9 |
| KMISPool | AAAI'23 | The independent sets $K$: 1, 2, 3, 4, 5 |
| GSAPool | WWW'20 | Pooling ratio: 0.1, 0.3, 0.5, 0.7, 0.9; Alpha: 0.2, 0.4, 0.6, 0.8 |
| HGPSLPool | Arxiv'19 | Pooling ratio: 0.1, 0.3, 0.5, 0.7, 0.9 |
| ParsPool | ICLR'24 | Parsingnet layers: 1, 2, 3; Deepsets layers: 1, 2, 3 |
| *Node Clustering Pooling* | | |
| AsymCheegerCutPool | ICML'23 | MLP layers: 1, 2; MLP hidden channels: 64, 128, 256 |
| DiffPool | NIPS'18 | Not applicable |
| MincutPool | ICML'20 | Temperature: 1, 1.5, 1.8, 2.0 |
| DMoNPool | JMLR'23 | Clusters: 2, 4, 6, 8, 10, 12 |
| HoscPool | CIKM'22 | Mu: 0.2, 0.5, 0.8; Alpha: 0.2, 0.5, 0.8 |
| JustBalancePool | Arxiv'22 | Not applicable |
| SEPool | ICML'22 | Tree depth: 1, 2, 3; Number of blocks: 1, 2, 3, 4 |

## B.2 Graph Regression

We use a backbone network inspired by MESPool (Xu et al., 2024a) for graph regression. The model mainly consists of three GINConv layers with ReLU activation functions and BatchNorm, along with two pooling layers, followed by a global average pooling layer. All channels (both hidden and output) are set to 64. The embedding output from the global average pooling layer passes through another linear layer with ReLU activation, having dimensions (64, 32). All models use the Adam optimizer with a learning rate of 0.001 and are trained for 200 epochs by minimizing the negative log-likelihood loss function. All data are processed using a 5-fold cross-validation and are run on multiple different seeds.

## B.3 Node Classification

For node classification, we utilize a U-Net architecture, which we divide into a downsampling convolutional part and an upsampling convolutional part (Siddique et al., 2021). The downsampling convolutional section includes two GCNConv layers with ReLU activation functions, with pooling applied between these layers. In the upsampling convolutional section, we use the indices saved during pooling for upsampling, restoring features to their pre-pooling size. The upsampled features are then fused with the corresponding residual features from the downsampling path, either through summation or concatenation. Finally, the fused features are processed and activated through a GCNConv layer. All models employ the Adam optimizer with a learning rate set to 0.001 and are trained for 200 epochs using cross-entropy loss. All data are processed using a 5-fold cross-validation and are run on multiple different seeds.

## B.4 Hyperparameter Tuning

Details of hyperparameter tuning for different pooling methods can be found in Table A.4. We performed hyperparameter searches for each dataset in each task.

Table A.5: Results of **node classification under real-world noise**.

| Dataset | TopKPool | SAGPool | ASAPool | PANPool | COPool | CGIPool | KMISPool | GSAPool | HGPSLPool |
|---|---|---|---|---|---|---|---|---|---|
| Cora | 74.16±0.29 | 74.00±0.59 | 74.52±0.08 | 74.09±0.32 | 73.10±0.22 | 73.67±0.05 | 74.59±0.27 | 74.24±0.25 | **74.73±0.23** |
| CiteSeer | 72.31±0.07 | 72.48±0.10 | 72.68±0.19 | 72.67±0.17 | 72.49±0.16 | 72.31±0.21 | **72.74±0.28** | 72.49±0.09 | 72.61±0.21 |
| PubMed | 78.65±0.11 | 78.57±0.06 | OOM | 77.71±0.05 | 78.43±0.17 | 78.35±0.04 | **78.84±0.09** | 78.68±0.09 | OOM |

Table A.6: Results of **node classification under label noise attack**.

| Dataset | TopKPool | SAGPool | ASAPool | PANPool | COPool | CGIPool | KMISPool | GSAPool | HGPSLPool |
|---|---|---|---|---|---|---|---|---|---|
| Cora | 61.16±0.11 | 61.16±0.23 | 61.60±0.08 | 61.56±0.27 | 60.65±0.14 | 60.86±0.28 | 61.62±0.16 | 61.21±0.15 | **61.64±0.17** |
| CiteSeer | 58.63±0.67 | 53.25±0.30 | 53.54±0.23 | 53.49±0.13 | 52.76±0.18 | 52.60±0.33 | 53.71±0.22 | 53.37±0.06 | 53.72±0.10 |
| PubMed | 62.42±0.09 | 62.37±0.08 | **62.50±0.11** | 62.22±0.04 | 62.14±0.08 | 62.02±0.16 | 62.28±0.07 | 62.47±0.08 | 62.16±0.07 |

## C  Additional Experiments

### C.1  Robustness Analysis

Table A.7 shows the additional robustness analysis on more node-level datasets. From Table A.7, we observe the following: Firstly, for smaller node classification datasets such as Cornell, Texas, and Wisconsin, masking node features results in the greatest performance loss, while edge deletion leads to the smallest performance loss. The potential reason is that these smaller datasets inherently possess higher local characteristics and structural sparsity, making node features more critical for the model's classification tasks. Secondly, consistent with the robustness analysis on Cora, Citeseer, and Pubmed, ASAPool and KMISPool demonstrate superior performance, indicating that these pooling methods exhibit stronger robustness in node classification tasks.

To generate real-world noise due to spurious or missing links, we follow current works (Xia et al., 2023; Zhao et al., 2024) to build a kNN graph instead of randomly adding noise. The performance can be found as in Table A.5. From the results, we can find that the differences among various pooling methods are relatively small, which aligns with the observations from noise attacks involving random addition/removal of edges and random masking of node features. Basically, KMISPool, GSAPool, and HGPSLPool demonstrate the best overall performance. We have also introduced the label noise by randomly modifying the class labels of 30% of the nodes. From the results in Table A.6, we can observe a similar observation.

### C.2  Generalizability Analysis

Table A.8 presents the results of size-based and density-based distribution shifts on NCI109 and IMDB-B, respectively. From Table A.8, we obtain conclusions similar to those in the main text: for the NCI109 dataset, node dropping pooling methods perform worse than node clustering pooling methods, whereas on the IMDB-B dataset, node dropping pooling methods outperform node clustering pooling methods. Overall, AsymCheegerCutPool, MinCutPool, and DMoNPool outperform other pooling methods.

Table A.9 presents the results of degree-based and closeness-based distribution shifts on node classification tasks across four datasets: Pubmed, Cornell, Texas, and Wisconsin. We observe the following: *Firstly*, KMISPool and GSAPool generally perform the best, yet no single pooling method consistently leads across all datasets. *Secondly*, the issue of class imbalance persists, and it is more pronounced in smaller datasets such as Cornell, Texas, and Wisconsin. *Thirdly*, smaller datasets like Cornell, Texas, and Wisconsin are more sensitive to distribution shifts compared to the larger dataset Pubmed, resulting in more significant performance degradation.

Table A.7: Results of **node classification under random noise attack**.

| Dataset | Ptb Method | TopKPool | SAGPool | ASAPool | PANPool | COPool | CGIPool | KMISPool | GSAPool | HGPSLPool |
|---------|-----------|----------|---------|---------|---------|--------|---------|----------|---------|-----------|
| Cornell | ADD | 46.64±0.25 | 46.45±0.77 | **47.18±1.03** | 47.00±1.18 | 46.63±0.25 | 46.63±0.92 | 46.81±0.67 | 46.81±1.12 | 46.45±0.45 |
| | DROP | 61.73±1.17 | 62.09±1.84 | 62.29±1.15 | 61.93±0.95 | 62.25±0.44 | 61.36±2.00 | **63.19±1.79** | 63.01±0.54 | 62.46±1.33 |
| | MASK | 46.99±0.90 | 47.36±1.56 | 47.91±2.46 | 47.00±0.45 | **48.11±1.17** | 46.63±1.03 | 47.74±1.13 | 47.01±1.35 | 47.55±1.19 |
| Texas | ADD | 58.63±0.67 | **61.37±0.94** | 60.48±0.62 | 59.92±0.92 | 59.01±0.44 | 58.99±1.14 | 58.83±0.49 | 59.37±1.39 | 58.63±0.23 |
| | DROP | 64.47±2.02 | 64.47±0.42 | 63.92±2.38 | 64.30±0.89 | 63.92±1.96 | 63.57±1.40 | 65.57±1.59 | **65.57±1.33** | 63.57±2.06 |
| | MASK | 57.56±0.92 | 57.93±1.55 | **58.85±1.44** | 57.19±0.91 | 57.37±1.33 | 57.56±0.93 | 58.10±1.09 | 57.93±0.92 | 57.92±0.43 |
| Wisconsin | ADD | 54.46±0.99 | 53.66±0.98 | **56.18±1.49** | 55.52±0.21 | 55.78±0.98 | 55.78±1.30 | 54.99±0.34 | 54.73±0.48 | 53.65±0.37 |
| | DROP | 61.23±1.23 | 60.55±1.15 | 60.29±1.68 | 59.76±1.72 | 60.43±1.79 | 59.36±0.87 | 60.16±0.66 | 60.43±0.95 | 61.36±1.17 |
| | MASK | 47.02±0.55 | 47.94±0.18 | 49.80±0.86 | 48.60±1.14 | **52.59±0.01** | 48.35±0.49 | 49.53±1.15 | 48.20±1.11 | 48.08±0.96 |

Table A.8: Results of **graph classification under distribution shifts**.

| Method | NCI109 | | | | IMDB-B | | | |
|--------|--------|--------|--------|--------|--------|--------|--------|--------|
| | Size | | Density | | Size | | Density | |
| | Micro-F1 | Macro-F1 | Micro-F1 | Macro-F1 | Micro-F1 | Macro-F1 | Micro-F1 | Macro-F1 |
| *Node Dropping Pooling* | | | | | | | | |
| TopKPool | 25.10±0,81 | 22.90±1.01 | 55.92±2.57 | 54.88±1.42 | 56.00±1.22 | 53.34±2.26 | 72.00±5.94 | 68.13±5.78 |
| SAGPool | 24.07±1.29 | 21.64±1.58 | 53.31±1.74 | 51.46±0.98 | 67.67±5.10 | 67.44±4.97 | 74.27±3.92 | 67.13±5.78 |
| ASAPool | 22.57±0.70 | 19.42±1.04 | 58.42±2.73 | 57.09±2.02 | 73.83±12.43 | 73.70±12.36 | **80.80±4.94** | **78.70±4.15** |
| PANPool | 25.73±0.11 | 23.80±0.12 | 56.25±1.63 | 54.34±1.93 | 66.50±9.34 | 65.73±10.38 | 76.27±4.21 | 71.65±4.67 |
| COPool | 24.94±1.72 | 22.77±2.31 | 57.10±2.20 | 55.39±1.21 | 65.33±2.72 | 65.13±2.94 | 72.40±9.91 | 69.52±8.26 |
| CGIPool | 24.54±1.19 | 22.39±1.46 | 61.36±0.84 | 57.98±3.78 | 72.83±8.00 | 72.69±7.87 | 71.60±5.44 | 63.60±7.27 |
| KMISPool | 43.78±5.82 | 43.24±5.33 | 58.08±3.45 | 50.03±6.64 | 75.33±4.78 | **75.16±4.61** | 78.80±0.86 | 73.74±0.63 |
| GSAPool | 25.97±3.11 | 24.00±4.03 | 53.04±1.30 | 52.64±1.16 | 70.17±3.70 | 69.17±4.45 | 78.80±0.86 | 73.11±0.99 |
| HGPSLPool | 21.54±0.22 | 18.17±0.43 | 58.18±2.26 | 55.06±3.02 | 69.33±4.71 | 69.25±4.76 | 75.07±1.86 | 70.37±1.81 |
| ParsPool | 42.68±0.81 | 41.97±0.74 | 59.91±1.13 | 56.13±2.70 | 75.00±3.63 | 74.92±3.57 | 76.00±2.99 | 71.63±3.00 |
| *Node Clustering Pooling* | | | | | | | | |
| AsymCheegerCutPool | 79.18±0.00 | 49.92±0.12 | 68.53±0.00 | 44.29±0.00 | 71.50±0.60 | 71.45±0.60 | 78.80±0.00 | 73.75±0.00 |
| DiffPool | 21.38±0.00 | 17.61±0.00 | 69.47±0.00 | 48.65±0.02 | 69.17±0.70 | 67.31±0.83 | 65.20±1.10 | 62.72±0.76 |
| MinCutPool | 31.83±0.77 | 30.50±0.90 | 70.76±0.00 | 56.27±0.02 | 70.17±0.01 | 68.28±0.03 | 78.40±0.01 | 73.91±0.02 |
| DMoNPool | **79.41±0.01** | **55.84±0.01** | 67.44±0.05 | 55.80±0.13 | 74.33±1.59 | 73.85±1.61 | 77.33±0.13 | 72.56±0.20 |
| HoscPool | 33.73±2.35 | 30.58±2.13 | 69.54±0.05 | 54.41±0.03 | 72.83±0.09 | 72.37±0.10 | 77.20±0.04 | 73.43±0.05 |
| JustBalancePool | 58.43±4.14 | 44.15±1.17 | **71.26±0.00** | **58.08±0.01** | **76.17±0.17** | 74.69±0.35 | 78.40±0.01 | 73.91±0.02 |

## C.3 Hyperparameter Sensitivity Analysis

Figure A.1 presents the results of a hyperparameter sensitivity analysis conducted on several pooling methods, evaluated on MUTAG and IMDB-MULTI. In particular, we conduct experiments on MUTAG and IMDB-MULTI by varying the number of hidden channels in $\{32, 64, 128, 256, 512\}$ and learning rates in $\{0.001, 0.005, 0.01, 0.05, 0.1\}$ for the pooling methods. From the results, we can find that the performance of pooling methods fluctuates with changes in hidden channels and learning rates. For smaller datasets such as MUTAG, hidden channels of 32 and 64 combined with smaller learning rates yielded better performance. For larger datasets such as IMDB-MULTI, hidden channels of 256 and slightly larger learning rates demonstrated superior performance.

## C.4 Visualization

Figure A.2 visualizes the first and second pooling steps of GSAPool, TopKPool, and SAGPool, as well as the original unpooled graph, on the third graph of the MUTAG dataset. For small-scale graphs such as MUTAG (Avg. nodes=17.9, Avg. edges=39.6), GSAPool partitions the graph into distinct "communities" during pooling, while TopKPool and SAGPool maintain graph connectivity. This demonstrates the superior performance of GSAPool, as it retains nodes from different semantic units (e.g., functional groups), whereas TopKPool and SAGPool tend to select nodes from a single dense region (e.g., the molecular backbone), potentially losing critical functional group information. This further validates the advantage of GSAPool, as its dual-modal filtering of structural information (SBTL) and node feature information (FBTL) accurately identifies community cores, enabling the classifier to directly capture chemical patterns. Figure A.3 and Figure A.4 illustrate the first and second pooling steps of KMISPool, TopKPool, and SAGPool, along with

Table A.9: Results of **node classification under distribution shifts**.

| Method | Pubmed | | | | Cornell | | | |
|---|---|---|---|---|---|---|---|---|
| | **Degree** | | **Closeness** | | **Degree** | | **Closeness** | |
| | **Micro-F1** | **Macro-F1** | **Micro-F1** | **Macro-F1** | **Micro-F1** | **Macro-F1** | **Micro-F1** | **Macro-F1** |
| TopKPool | 81.66±0.32 | 81.11±0.38 | 85.66±0.09 | 82.36±0.24 | 34.07±2.10 | 22.35±2.92 | 57.78±4.80 | 25.56±8.37 |
| SAGPool | 83.07±1.02 | 82.57±1.07 | **85.96±0.34** | **82.73±0.38** | **36.30±4.19** | 24.39±4.03 | 61.48±4.57 | **31.54±8.24** |
| ASAPool | 82.11±0.12 | 81.94±0.55 | 85.02±0.54 | 82.12±0.77 | 34.07±1.05 | 20.94±3.40 | 60.74±2.77 | 26.33±2.21 |
| PANPool | 81.65±0.12 | 81.15±0.11 | 85.55±0.03 | 82.16±0.20 | 35.56±0.00 | 23.55±0.90 | 54.07±2.10 | 18.58±3.28 |
| COPool | 80.42±0.14 | 79.82±0.20 | 84.99±0.35 | 81.81±0.29 | 35.56±1.81 | 22.64±3.21 | 49.63±6.87 | 21.09±3.44 |
| CGIPool | 80.35±1.33 | 79.69±1.32 | 84.97±0.75 | 81.74±0.90 | **36.30±4.19** | **25.34±4.15** | 60.00±1.81 | 27.48±3.60 |
| KMISPool | **83.41±0.02** | **82.92±0.03** | 85.66±0.07 | 82.50±0.12 | 34.81±1.05 | 22.99±2.00 | 58.52±5.54 | 24.70±8.45 |
| GSAPool | 83.15±0.81 | 82.61±0.83 | 85.83±0.32 | 82.70±0.50 | 35.56±1.81 | 23.23±1.21 | **62.22±1.81** | 30.06±4.20 |
| HGPSLPool | OOM | OOM | OOM | OOM | 34.07±1.05 | 21.39±1.41 | 57.78±1.81 | 23.63±1.61 |

| Method | Texas | | | | Wisconsin | | | |
|---|---|---|---|---|---|---|---|---|
| | **Degree** | | **Closeness** | | **Degree** | | **Closeness** | |
| | **Micro-F1** | **Macro-F1** | **Micro-F1** | **Macro-F1** | **Micro-F1** | **Macro-F1** | **Micro-F1** | **Macro-F1** |
| TopKPool | **40.74±2.10** | **27.06±2.71** | 61.48±1.05 | 19.39±0.07 | 26.88±1.52 | 19.88±1.97 | 17.74±2.28 | 15.34±1.55 |
| SAGPool | 39.26±1.05 | 23.52±0.35 | 62.22±1.81 | 19.44±0.35 | 27.42±1.32 | **25.32±0.98** | 18.28±0.76 | 14.85±0.25 |
| ASAPool | 40.74±2.77 | 23.87±0.67 | 62.96±1.05 | 19.58±0.20 | 29.03±3.48 | 20.17±1.42 | 15.59±2.74 | 11.63±4.27 |
| PANPool | 40.00±3.63 | 23.64±1.13 | 63.70±1.05 | 19.72±0.20 | 30.11±3.31 | 22.99±1.60 | 23.66±10.56 | 15.42±5.61 |
| COPool | 39.26±1.05 | 23.44±0.32 | 63.70±2.10 | 19.72±0.39 | 28.49±1.52 | 19.94±1.50 | 16.13±3.95 | 16.68±3.84 |
| CGIPool | 40.00±1.81 | 23.85±0.52 | 60.74±1.05 | 19.25±0.18 | 25.27±2.74 | 20.01±2.23 | **27.96±19.01** | **20.37±11.58** |
| KMISPool | 38.52±1.05 | 23.19±0.10 | **63.70±1.05** | **19.72±0.20** | **31.72±4.23** | 22.58±2.76 | 16.67±0.76 | 14.20±0.97 |
| GSAPool | **40.74±2.10** | 26.90±2.72 | 62.22±0.00 | 19.44±0.00 | 26.88±3.31 | 18.51±1.18 | 15.59±1.52 | 12.75±1.41 |
| HGPSLPool | 40.74±2.77 | 25.69±3.19 | 62.96±1.05 | 19.58±0.20 | 30.65±1.32 | 20.76±1.23 | 16.13±2.63 | 12.67±1.80 |

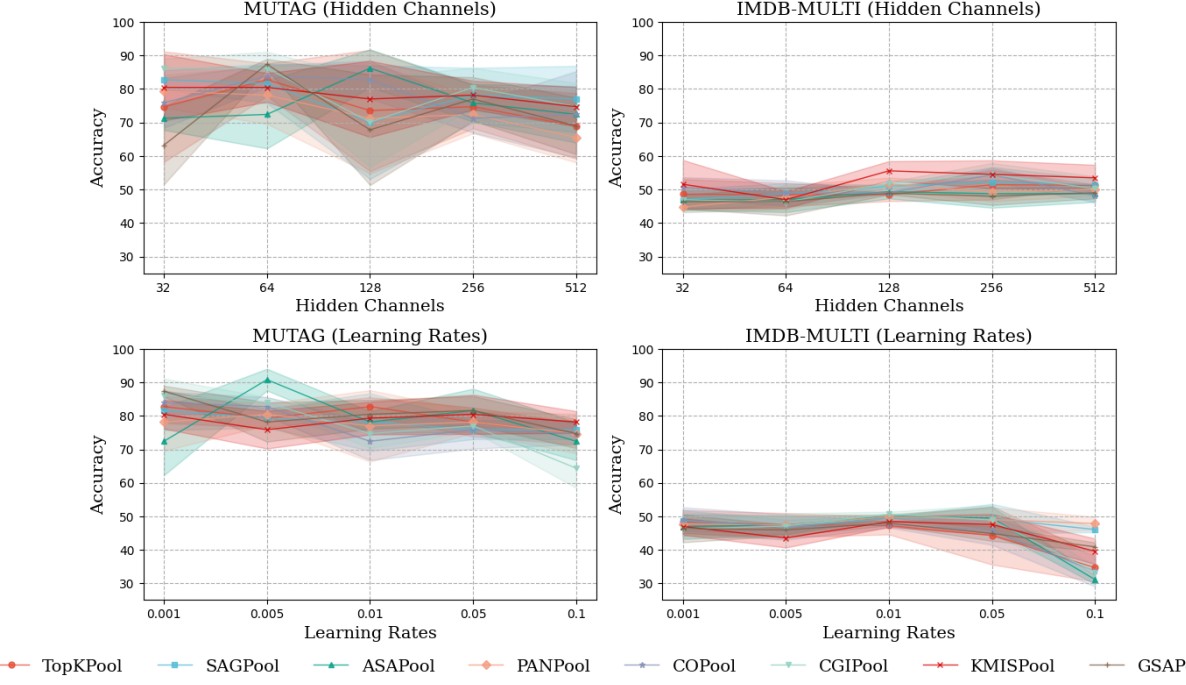

Figure A.1: Performance of different approaches under different number of hidden channels and learning rates.

the original unpooled graph, on the Texas and Cora datasets. For datasets with weaker connectivity, such as Texas (Avg. degree=3.2), and those with higher clustering coefficients, such as Cora (Avg. CC=0.24), KMISPool retains more central nodes and preserves connectivity more effectively. In contrast, SAGPool and TopKPool retain fewer nodes and often fail to preserve high-degree nodes. SAGPool and TopKPool rely

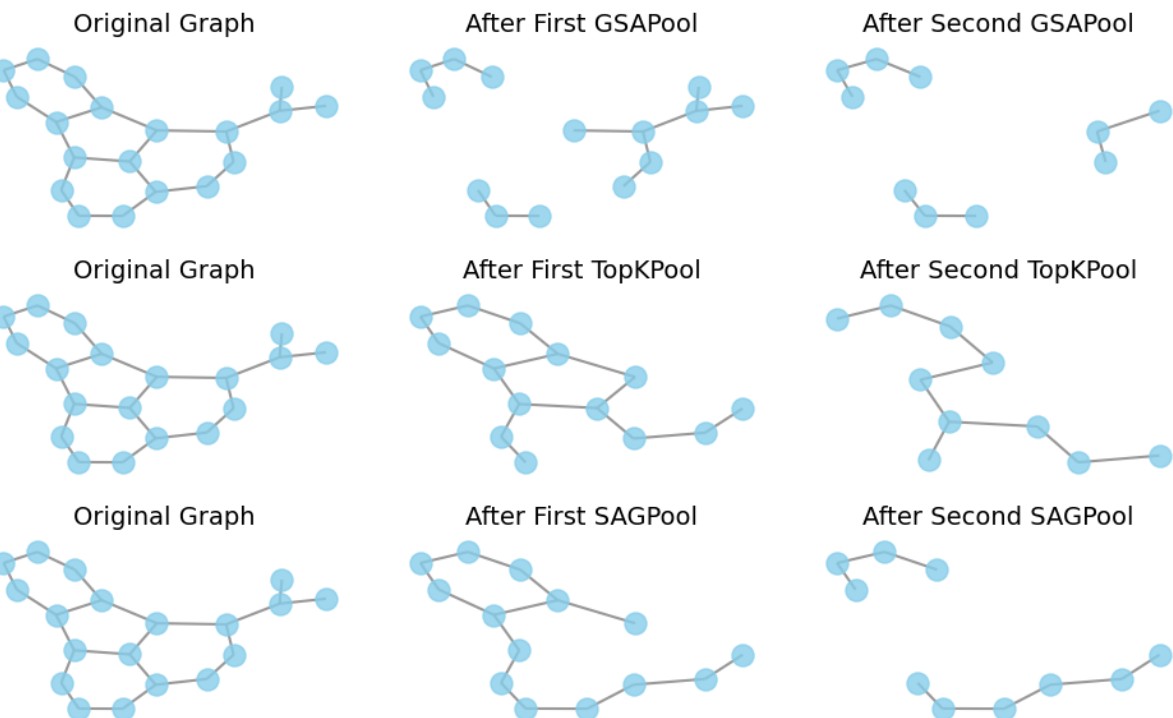

Figure A.2: Visualization of different approaches for one graph of MUTAG.

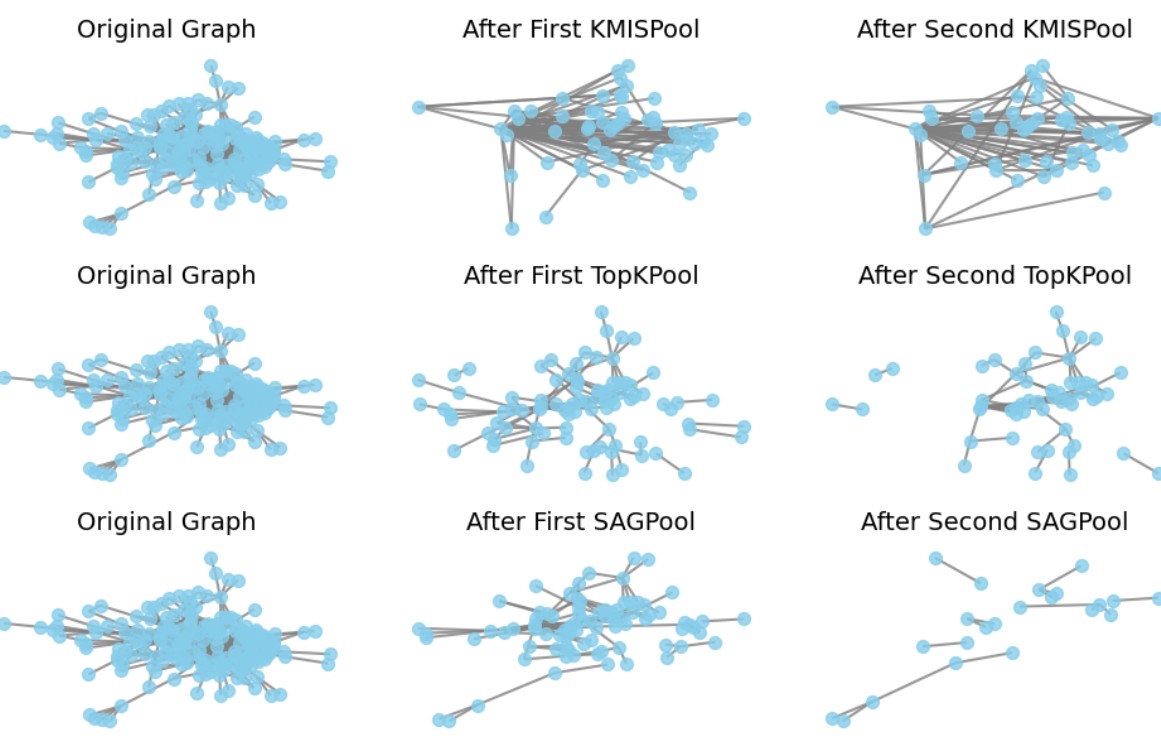

Figure A.3: Visualization of different approaches for Texas.

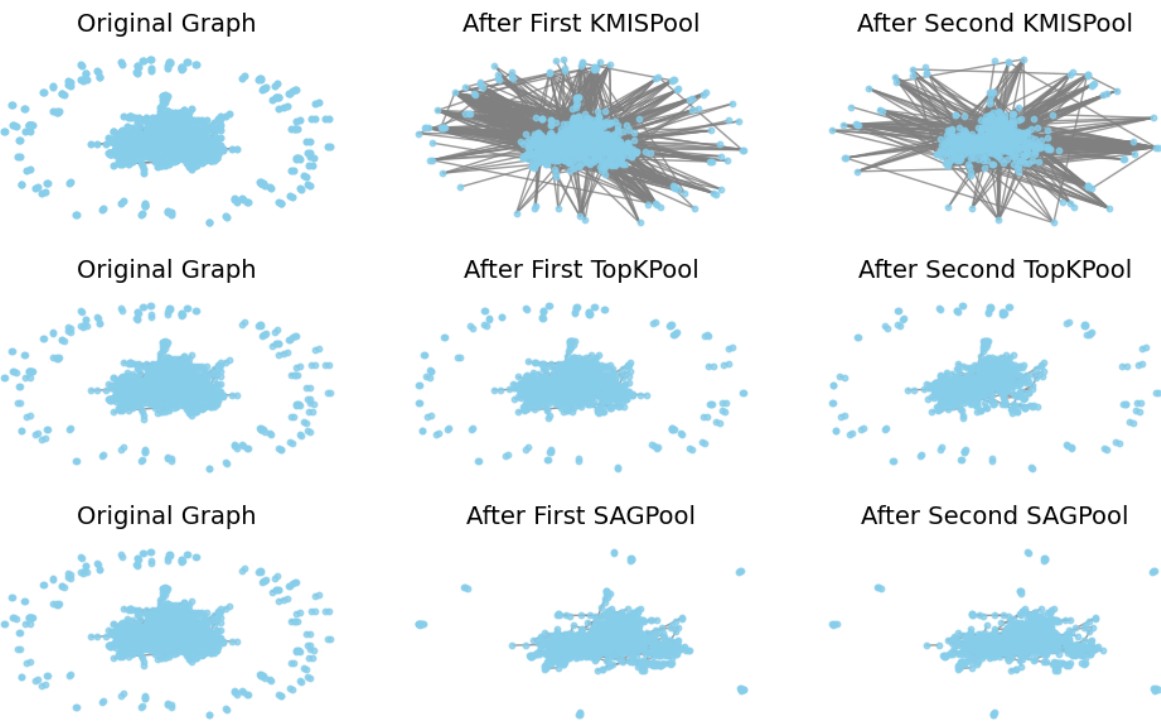

Figure A.4: Visualization of different approaches for Cora.

heavily on local features, such as node degrees or attention scores, for node selection. In weakly connected graphs, this local strategy may lead to the omission of high-degree nodes, as they may be concentrated in specific regions, while other regions are oversampled. In comparison, KMISPool achieves more uniform sampling by leveraging its global node selection strategy.

