# OpenReview forum: "A Comprehensive Graph Pooling Benchmark: Effectiveness, Robustness and Generalizability"
_TMLR — Withdrawn by Authors_

### Review · Reviewer_2pWk · 2025-01-03

**Summary Of Contributions:**

This paper conducts the experimental study about the 17 graph pooling methods on 28 graph datasets. The author varies the pooling layer and evaluates the methods by the overall graph and node classification and regression performance. Further, the author evaluates several aspects such as the robustness, generalization performance and computational cost.

**Audience:**

Yes

**Broader Impact Concerns:**

It is a benchmark evaluation paper and methods and resources are from existing work, I think the potential ethical problem is small.

**Claims And Evidence:**

Yes

**Requested Changes:**

Discussion on the motivation of focusing on the pooling layer will make the impact of the paper larger.

Hyperparameter sensitivity analysis will make the results more reliable.

The rest two weakness comments are minor and for the clarity.

**Strengths And Weaknesses:**

Strength

The author conducts extensive experimental study about the pooling methods. Multiple methods are conducted on several benchmark datasets from wide aspects. These results would help the researcher of this field.


Weakness

Since the graph neural networks consists of several layers including pooling layer, it would be better to add more discussion why the author focuses on the pooling layer.

As written in B.4, pooling methods have hyperparameters. More discussion of the hyper parameter tuning method would be helpful (e.g. cross validation). Further, it would be better to add hyperparameter sensitivity analysis.

In Figure 2, there are arrows labeled with ‘Pooling’ that increases the number of nodes, which contradicts to the definition of the pooling layer that |V’| < |V|. Are unpooling or upsampling appropriate?

Since the experiment section is long and there are various experiments conducted, it would help reader to add the section or paragraph that summarizes the finding of the experimental results.

---

### Review · Reviewer_CqxU · 2025-01-06

**Summary Of Contributions:**

The paper presents a benchmark of graph pooling strategy in graph neural networks analyzing 17 methods for graph pooling and 28 datasets.  The paper presents analysis of the results of executing these methods based on the effectiveness, robustness, and generalizability of the methods.  The paper concludes that node clustering pooling outperform node dropping methods.  The paper makes no claims about the models and limits to show the experimental results and facts about the experiments.

**Audience:**

Yes

**Broader Impact Concerns:**

The paper analyzes graph neural networks' pooling mechanism in several datasets.  There is no concerns on my part from the content presented in the paper.

**Claims And Evidence:**

Yes

**Requested Changes:**

Major:
- In Fig. 3, when discussing the noise intensity, did the authors consider noise in the edges due to spurious or missing links instead of just adding noise?  This other type of noise edges should be discussed and analyzed if possible.

Minor:
- Typo in P4 "($H^{(l+1)}"
- Normalize the y-axis on all plots on Fig. 3
- Leftover sentence in P10 "As depicted in"

**Strengths And Weaknesses:**

**Strengths:**
- The paper benchmarks several methods for graph pooling on a wide set of datasets.
- The paper shows analysis of the benchmarked methods according to the effectiveness, robustness, and generalizability.
- The paper found that node clustering outperforms node dropping based on the robustness and generalizability but have higher computational cost.

**Weaknesses:**
- The conclusions are limited to changing the graph pooling mechanism in the models.  While I understand that testing other parts of the methods will be expensive, the results and conclusions could be more meaningful.
- The knowledge gain from the paper is limited given that it presents the execution of methods, with limited analysis and conclusions.

---

> ### Comment · Reviewer_CqxU · 2025-01-24
> **Thanks for the update**
>
> I thank the authors for the updated text.  Their updated paper addressed the raised changes about the experiments and minor changes.
>
> The weaknesses regarding the limited knowledge gain still remains since it is a core feature of the paper structure.

---

### Review · Reviewer_fLXh · 2025-01-09

**Summary Of Contributions:**

This paper presents a **comprehensive benchmark** for evaluating graph pooling methods, addressing the lack of standardized experimental settings in this domain. The benchmark incorporates **17 graph pooling approaches** and **28 datasets**, assessing their performance across three key dimensions: **effectiveness, robustness**, and **generalizability**.

**Audience:**

Yes

**Claims And Evidence:**

Yes

**Requested Changes:**

Please see the weakness part.

**Strengths And Weaknesses:**

Strengths:

* The inclusion of 17 graph pooling methods and 28 diverse datasets makes this benchmark a significant resource for evaluating graph pooling methods in a standardized manner.
* The paper evaluates methods under challenging scenarios, such as OOD shifts and noise attacks, providing valuable observations.

Weaknesses:

* Although the paper provides detailed experimental observations, it lacks in-depth **insights** or interpretations. More discussion on why certain methods outperform others under specific conditions (e.g., settings, datasets) would strengthen the impact.
* The introduction and discussion of graph pooling techniques are somewhat superficial. For example, the pooling operations on graph structure (e.g., how pooling alters graph topology and connectivity) is just briefly mentioned.
*  The provided codebase, while useful, appears disorganized and lacks a unified structure. This makes it difficult for users to integrate or extend the benchmark efficiently. If the authors could provide a standardized API for accessing different pooling methods and datasets, it would significantly improve the usability and accessibility of the benchmark and increase its contribution to the community.

---

### Note · Authors · 2025-04-07

I have read and agree with the venue's withdrawal policy on behalf of myself and my co-authors.